# Standardization of ELISA protocols for serosurveys of the SARS-CoV-2 pandemic using clinical and at-home blood sampling

Carleen Klumpp-Thomas [1,2,7], Heather Kalish[3,7], Matthew Drew[4], Sally Hunsberger[5], Kelly Snead[4], Michael P. Fay [5], Jennifer Mehalko[4], Anandakumar Shunmugavel[2], Vanessa Wall[4], Peter Frank[4], John-Paul Denson[4], Min Hong[4], Gulcin Gulten[4], Simon Messing[4], Jennifer Hicks[3], Sam Michael[1], William Gillette[4], Matthew D. Hall[1], Matthew J. Memoli [6], Dominic Esposito[4] & Kaitlyn Sadtler [2✉]

The extent of SARS-CoV-2 infection throughout the United States population is currently unknown. High quality serology is key to avoiding medically costly diagnostic errors, as well as to assuring properly informed public health decisions. Here, we present an optimized ELISA-based serology protocol, from antigen production to data analyses, that helps define thresholds for IgG and IgM seropositivity with high specificities. Validation of this protocol is performed using traditionally collected serum as well as dried blood on mail-in blood sampling kits. Archival (pre-2019) samples are used as negative controls, and convalescent, PCR-diagnosed COVID-19 patient samples serve as positive controls. Using this protocol, minimal cross-reactivity is observed for the spike proteins of MERS, SARS1, OC43 and HKU1 viruses, and no cross reactivity is observed with anti-influenza A H1N1 HAI. Our protocol may thus help provide standardized, population-based data on the extent of SARS-CoV-2 seropositivity, immunity and infection.

[1] National Center for Advancing Translational Sciences, National Institutes of Health, Rockville, MD 20850, USA. [2] Section on Immuno-Engineering, National Institute of Biomedical Imaging and Bioengineering, National Institutes of Health, Bethesda, MD 20894, USA. [3] Trans-NIH Shared Resource on Biomedical Engineering and Physical Science, National Institute of Biomedical Imaging and Bioengineering, National Institutes of Health, Bethesda, MD 20894, USA. [4] Protein Expression Laboratory, NCI RAS Initiative, Frederick National Laboratory for Cancer Research, Frederick, MD 21702, USA. [5] Biostatistics Research Branch, National Institute of Allergy and Infectious Diseases, National Institutes of Health, Bethesda, MD 20894, USA. [6] LID Clinical Studies Unit, Laboratory of Infectious Diseases, Division of Intramural Research, National Institute of Allergy and Infectious Diseases, National Institutes of Health, Bethesda, MD 20894, USA. [7] These authors contributed equally: Carleen Klumpp-Thomas, Heather Kalish. ✉email: kaitlyn.sadtler@nih.gov

SARS-CoV-2 has spread across the globe rapidly, causing a worldwide pandemic[1]. Infection with this highly contagious respiratory virus can be asymptomatic or present as COVID19, a disease with varying levels of severity and a broad range of not fully understood symptoms that may include fever, cough, anosmia, gastrointestinal symptoms, hypercoagulability, inflammatory complications, acute respiratory distress syndrome, which may also lead to death[2–5]. Owing to the rapidly evolving nature of pandemics, the true extent of spread of SARS-CoV-2 will likely not be fully realized until late in–or even after—the pandemic. Moreover, as observed in all respiratory viral pandemics since 1918, the true number of infections always exceeds the detected cases[6,7]. In order to determine a better estimate of the prevalence of SARS-CoV-2 infection, high-quality serology assays must be developed. These assays measure the presence of antibodies against specific proteins of this novel coronavirus (to determine whether an individual has been infected with SARS-CoV-2) and aim for high sensitivity and specificity[8,9]. Both are important factors to diagnose prior infection; however, if a tradeoff between sensitivity and specificity is needed, high specificity should be emphasized when determining the extent of exposure across a population or for diagnosing previous infections. If such a highly specific, high-quality assay is available, then data can be generated from serosurveys and clinical testing that can be used to better understand the spread of infection, immunity, and correlates of protection.

Owing to the complex nature of immune responses and the temporal changes associated with canonical responses to infection[10–12], a combination of proper technical validity and proper interpretation of results is critical for understanding the meaning of the data acquired during serostudies. Previous studies have shown differing dynamics of antibodies, suggesting an important need for utilization of multiple antigens or multiple assays to properly measure seropositivity. More specifically, a degradation of anti-nucleocapsid titers has been reported, whereas anti-Spike antibodies appear to persist for a longer duration[13,14].

Furthermore, certain specific antibodies, such as those against the receptor-binding domain (RBD) of SARS-CoV-2 spike, can correlate well with neutralization[15], which has also been shown for the original SARS-CoV[16], though it may miss a polyclonal response as non-RBD binding neutralizing antibodies have been reported[17,18].

Here, we present an optimized enzyme-linked immunosorbent assays (ELISA)-based serology assay protocol—from protein production to data analysis—that analyzes the presence of IgG, IgM, and IgA antibodies against spike and RBD antigens of SARS-CoV-2. We also demonstrate how our testing protocol can be validated and how specific thresholds for positivity can be set for manual and semi-automated methods. Evaluation of all parameters of serologic assays, such as those in the workflow presented here, is critical for proper interpretation of antibody testing both in clinical and public health applications.

## Results

In order to properly prepare to generate such useful data from a now-ongoing National Institutes of Health (NIH) sponsored national serosurvey in the United States (NCT04334954), we developed a serology protocol that emphasizes specificity while maintaining a simple approach that can be repeated at relatively low cost in laboratories without specialized equipment. The NIH serosurvey study allows mail-in home sampling using dried blood on a microsampler or collection of blood on-site. Therefore, we developed, implemented, and evaluated a serology testing protocol using ELISA that could successfully be used with multiple sample types, while emphasizing the specificity required to conduct high-quality convalescent testing and serosurveys (Fig. 1).

**Optimization of protein production and purification.** The CoV-2 spike protein RBD was produced by secretion from Expi293 cells at high yield (>15 mg/l) after 72 hours of expression. Little improvement in yield was seen with extended-expression

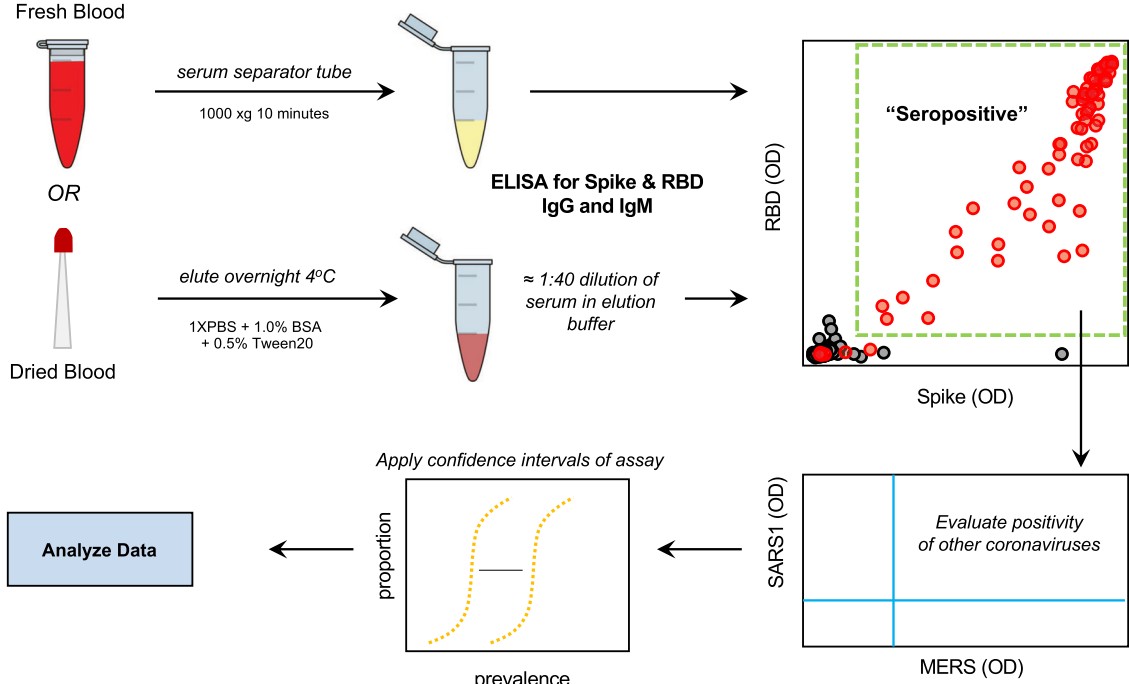

**Fig. 1 Serology testing protocol for evaluation of SARS-CoV-2 seropositivity in a large-scale population.** Utilizing both venipuncture-derived fresh blood and dried blood spots, we have standardized a dual-antigen ELISA platform for highly specific (IgG = 100%, 95% confidence interval = 98.5–100) detection of SARS-CoV-2 antibodies for application in precise, large-scale serosurvey efforts.

times. Although the Mt. Sinai and Ragon constructs differ significantly in their C-terminal tags and use different secretion signal peptides (native spike for Mt. Sinai, tissue plasminogen activator for Ragon), the behavior of these proteins during purification was highly similar. SARS-CoV-2 spike proteins, by contrast, were poorly secreted with yields of no >2 mg/l in our hands, similar to reports in the literature[19,20]. Improved production was seen at 96 hours versus 72 hours of expression; however, viabilities of cultures were very low after 96 hours, suggesting toxicity perhaps owing to failures in the secretory pathway or activation of the unfolded protein response. Curiously, expression of other coronavirus spike proteins (SARS1, MERS, OC43, HKU1) consistently produced higher yields (5–11 mg/l) compared with CoV-2 for reasons that are still unclear. Purification of spike proteins was further complicated by significant protein loss during size-exclusion chromatography (SEC). Ultimately, we chose to eliminate the SEC step and instead modified the immobilized metal affinity chromatography (IMAC) purification conditions to produce higher purity material, followed by desalting to the final buffer. This eliminated the protein loss during SEC and produced proteins that were similar in purity to those seen with the original protocol. Figure 2b shows the final purified proteins generated using the modified protocols. To verify that spike proteins formed the expected trimeric structures, analytical SE was performed (Fig. 2c) and McLellan/VRC and

Mt. Sinai spike proteins showed a clear elution peak corresponding to the size of a trimer, with no apparent monomeric peaks. In addition, transmission electron microscopy images (Fig. 2c, inset) confirmed the presence of particles similar in size and appearance to previously characterized CoV-2 trimers[20]. Taken together, these data suggest that the highly pure spike proteins are properly folded into their expected trimeric structures. Additional optimization of spike protein production was later carried out by lowering the expression temperature to 32 °C, which further improved yields of the protein to >5 mg/l. Under these optimal conditions, the McLellan/VRC construct had improved yield and purity when compared with the Mt. Sinai construct[21].

**Technical performance of multiple antigen constructs.** In order to determine the optimal assays for use in our serology protocol for SARS-CoV-2, we compared multiple antigens currently in use. These constructs included the spike ectodomain constructs from the McLellan Lab/NIAID VRC and from Florian Krammer's laboratory at Mount Sinai, as well as one RBD construct from the Ragon Institute of MGH, MIT, and Harvard, and one from the Krammer Lab of Mount Sinai[19,20] (Fig. 2d). Both spike constructs behaved similarly and showed no difference in ELISA signal across a range of recombinant antibody dilutions in serum

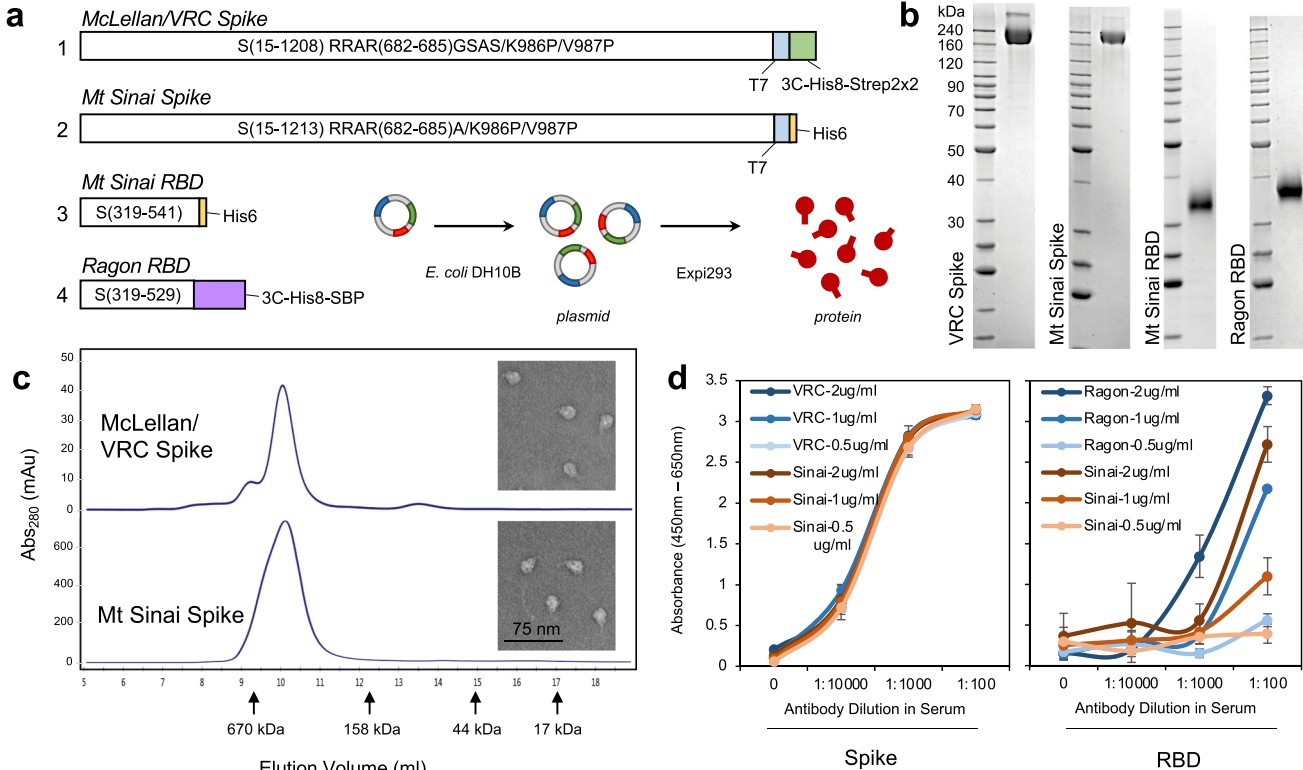

**Fig. 2 Production and sensitivity of two full spike ectodomain (Spike) and two receptor-binding domain (RBD) constructs used as antigens in ELISA. a** Schematic of the spike and RBD constructs used to generate recombinant proteins. Abbreviations are 3 C, rhinovirus 3 C protease cleavage site; *Strep2x2* dual Strep2 epitope tag, *T7* bacteriophage T7 fibritin trimerization domain, *SBP* streptavidin-binding peptide. **b** SDS-PAGE Coomassie Blue staining of purified (1) McLellan/VRC spike, n = 27, (2) Mt Sinai spike, n = 6, (3) Mt Sinai RBD, n = 5, and (4) Ragon RBD, n = 13, proteins. **c** Analytical size-exclusion chromatography of purified McLellan/VRC (n = 27) and Mt Sinai spike (n = 6) proteins. Peak elution volumes of sizing standards are noted (670 kDa—thyroglobulin, 158 kDa—gamma-globulin, 44 kDa—ovalbumin, 17 kDa—myoglobin). Inset: transmission electron microscopy of McLellan/VRC and Mt Sinai spike trimers. Ladder unit = kDa. **d** Left: full spike ectodomain at three different concentrations of protein coating density for both McLellan/VRC (NIAID Vaccine Research Center, blue) and Sinai (Mount Sinai, orange) constructs. Right: RBD constructs at three different concentrations of protein coating for both Ragon (Ragon Institute, blue) and Sinai (Mount Sinai, orange) constructs. Anti-spike or anti-RBD monoclonal recombinant antibody spiked into negative serum at 1:100, 1:1000, and 1:10000 dilution. Data are means ± SD, n = 4 independent study replicates on two independent protein preparations. Source data are provided as a Source Data file.

and blood that was loaded onto microsamplers. They also displayed similar sensitivity at multiple coating densities from 0.5 to 2 µg/ml. The two RBD constructs were similarly compared and displayed significantly different sensitivity at a range of recombinant antibody concentrations and coating densities. The RBD construct from the Ragon Institute displayed stronger signal with a recombinant RBD antibody when compared with the Krammer Lab construct. Therefore, based on these data we chose to use the McLellan/VRC spike construct and Ragon Institute RBD for our serological protocol by testing samples for the presence of antibody against both constructs.

**Sensitivity and specificity of selected ELISA assays and final protocol.** To evaluate specificity of the manual ELISA method, 100 negative controls (serum collected "pre-2019" and therefore prior to the emergence of SARS-CoV-2, Fig. 3) were assessed. At a 1:100 dilution of serum in the SARS-CoV-2 spike ELISA, there

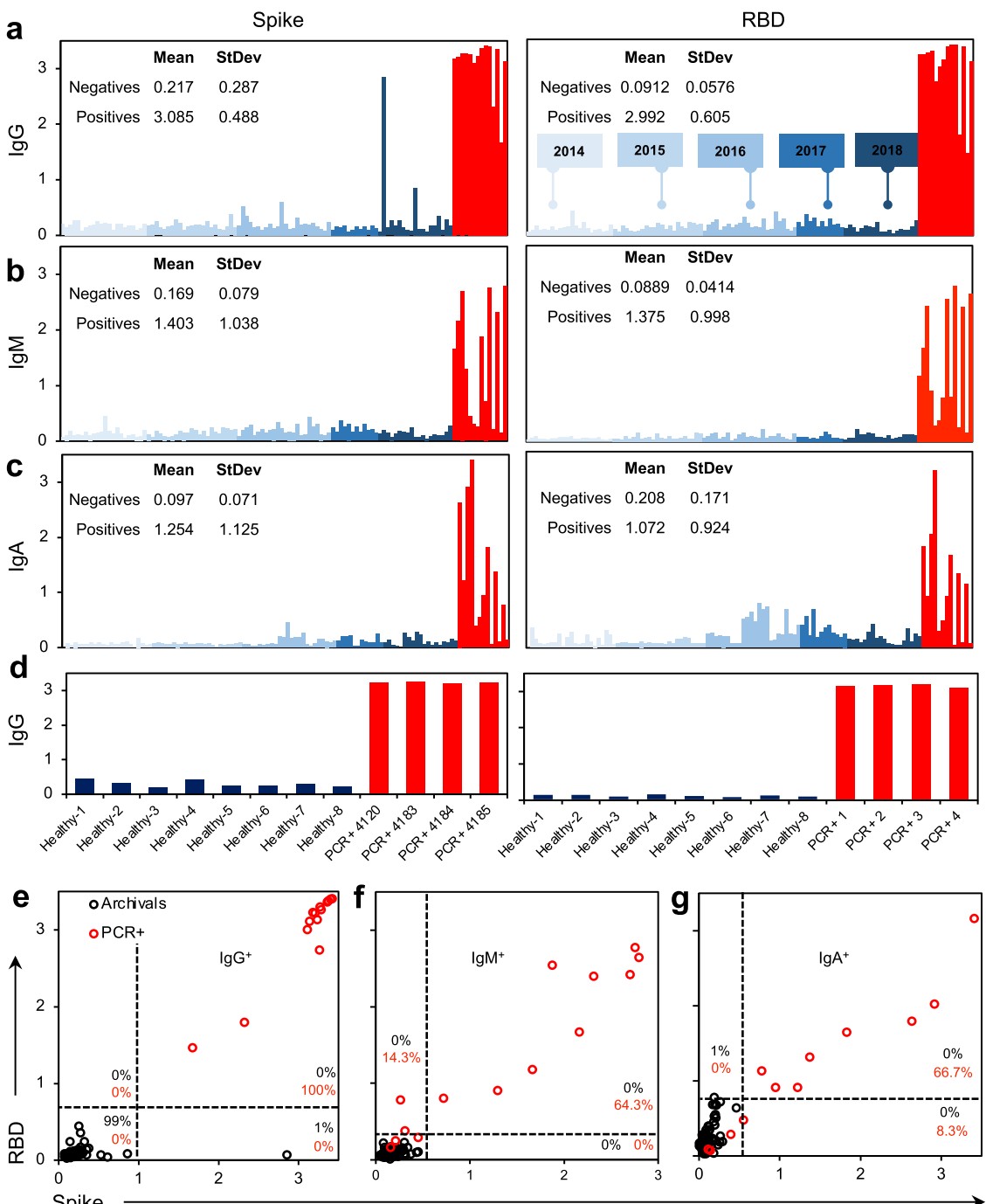

**Fig. 3 Specificity of antigens against a preliminary panel of pre-2019 archival sample controls. a–c** Spike (McLellan/VRC) and RBD (Ragon) antigens tested against 100 archival controls collected from 2014 to 2018 (blue) compared with 14 SARS-CoV-2 PCR+ controls (red) for **a** IgG, **b** IgM, and **c** IgA. **d** Microsampler controls of PCR+ samples against healthy controls. **e–g** Seropositivity thresholding for **e** IgG, **f** IgM), and **g** IgA calculated from average signal intensity of archival controls. n = 100 archival controls (black), n = 14 nasal swab SARS-CoV-2 PCR+ patients (red). Manual ELISA, titer = 1:400 (serum) or 1:10 (microsampler eluate). Source data are provided as a Source Data file.

was an unacceptable amount of background signal from the negative controls (Supplementary Fig. 1). At a 1:400 dilution, no loss of signal was observed from the positive controls and background was significantly diminished when assessing the negatives. A threshold of two standard deviations above the mean of the negative controls provided an estimated 98% specificity on the spike assay, whereas at a threshold of three standard deviations above the mean (OD = 1.078), an estimated 99% specificity was observed (Fig. 3). We repeated this assay for both IgM (OD = 0.405) and IgA (OD = 0.308), which resulted in estimated 100 and 99% specificity at three standard deviations above the mean. This was repeated with the RBD construct, where we observed 100% specificity for IgG (0.264), IgM (OD = 0.213), and IgA (OD = 0.723) at three standard deviations above the mean of the negative controls. Based on these data we propose using a threshold of three standard deviations above the mean of the negative controls for both the spike and RBD manual ELISA assays. Therefore, the definition of seropositivity in our protocol requires positivity (above the threshold) of both spike IgG and RBD IgG or both spike IgM and RBD IgM. This definition provides a specificity of 100% (95% CI 96.4%, 100%) from our negative controls using the manual ELISA method.

To increase our throughput and scale our studies, we moved toward a robotic high-throughput semi-automated platform. In addition to increasing the speed of sample processing, automation also increases standardization, minimizes technical variability, and keeps protocols consistent over long periods of time and multiple operators. As such, we implemented a parallel plate washer/dispenser in line with automated plate stackers and plate reader (Supplementary Fig. 2). Utilization of this platform minimized well-to-well and plate-to-plate technical variability (Supplementary Fig. 3). As automated systems alter the conditions and timing and do not precisely replicate manual pipetting, we re-titered known convalescent PCR + patient positive controls ($n = 9$) and archival negative controls ($n = 8$), representing the range of signal intensity (min to max) observed with both positives and negatives detected by the manual ELISA (Supplementary Fig. 4). Given the biologic difference in the concentrations of IgG, IgM and IgA, and based on the balance between background signal and positive signal, we found that a 1:400 dilution for IgG, IgM, and IgA results in clear signal with high positives, with multiple convalescent samples reaching the upper limit of detection of our instruments. We re-validated these titers on seropositive and seronegative identified donors from our small-scale test cohort using microsamplers loaded with donor blood then eluted as per standard protocol (Supplementary Fig. 5). To ensure equal endpoint serum dilution, we used an equivalent dilution of microsampler eluate for (1:10) equal to a 1:400 serum dilution. To re-establish thresholds and sensitivity on our automated setup that will be used in future serosurvey work, we evaluated the robotic setup against 300 archival pre-2019 negative controls (Supplementary Figs. 6, 7). At a threshold of the mean of the archival controls plus two standard deviations we detect 0 false positives for IgG (Supplementary Fig. 7a, d, thresholds: spike = 0.674, RBD = 0.306) and 0 false positives for IgM (Supplementary Fig. 7b, e, thresholds: spike = 0.208, RBD = 0.270). Using a threshold of three standard deviations above the mean, we detected 0 false positives for IgA detection (Supplementary Fig. 7c, f, thresholds: spike = 0.117, RBD = 0.148). Therefore, the preliminary specificity via the semi-automated system for IgG, IgM, and IgA was 100% (95% CI 98.78, 100.0). Therefore this semi-automated method gave us the same specificity we observed with the manual method, but with tighter confidence intervals demonstrating how with the accrual of further negative controls this method can be fine-tuned to thresholds to ensure very high specificity and, ultimately, with a

95% lower confidence limit of greater than 99%. We are utilizing IgM and IgG to threshold seropositivity as early infection could be signaled with IgM+ but IgG- donors; however, we expect the majority of seropositive individuals to be IgG+.

Sensitivity was evaluated first using the manual ELISA method on 14 positive control samples (i.e., serum collected from convalescent patients previously confirmed by PCR diagnostic testing to have had SARS-CoV-2). Samples were tested at the optimal 1:400 dilution observed for manual ELISA. Using our previously defined thresholds of three standard deviations above the mean of the negative controls for both spike and RBD, all positive control samples met the criteria for positivity. Therefore, the preliminary estimate of sensitivity was 100% (95% CI 76.8%, 100%) based on a small sample size. Once again, we also evaluated sensitivity of the semi-automated setup using our definitions by using 46 positive controls (Supplementary Fig. 7g). With the increased number of controls in the semi-automated method we observed 100% sensitivity with a tighter 95% confidence interval of 92.9–100%, again demonstrating how added controls will allow for further tightening of this confidence interval and a more accurate measure of sensitivity.

In addition to evaluating both the manual and semi-automated ELISA methods on serum, we also evaluated the performance of these assays on blood loaded onto the microsamplers, dried, and stored prior to elution. We did this by comparing 68 serum and microsamplers from the same donors. Antibodies eluted reliably from the microsamplers and were detected. Subsequently, antibody titers were performed and for IgG and IgM antibodies from positive control blood. There was a strong linear correlation between the serum and microsampler results (spike = 0.991, RBD = 0.961), demonstrating that our method worked well with both sample types. This suggests that not only could detection of antibodies be performed reliably, but quantification of antibodies from mail-in sampling devices was possible (Supplementary Fig. 8). Though we have displayed data on both serum and dried blood microsamplers (Neoteryx), any adaptation of this protocol should evaluate each sample source to ensure proper measurements of antibodies in blood or other body fluids.

**Results from a small-scale test sample set.** In order to further evaluate this serology testing protocol, we analyzed a set of samples from a high-exposure community. The PCR status and symptom status of the individuals were not linked to the samples, but all 68 donors were known to have had exposure to COVID19, 22 of whom had also tested positive in the recent past for SARS-CoV-2 infection by PCR. Using our testing protocol, we identified 59 individuals (from the total of 68) who met our criteria for positivity with both spike and RBD above the threshold for IgG and IgM. Subdividing the results based on the individual assays themselves, 59 tested positive for spike IgG, 31 tested positive for spike IgM, 62 tested positive for RBD IgG, and 51 tested positive for RBD IgM (Fig. 4a–d). When directly comparing the OD of spike versus RBD, we see that the regression fit for spike and RBD was not perfect for IgG ($R^2 = 0.856$) and IgM ($R^2 = 0.961$) (Fig. 4b, d, Supplementary Fig. 9). In particular, the $R^2$ for IgG was less than for IgM, and several points fell below the regression line when plotting spike as a function of RBD, suggesting that using spike alone may not be adequate. Measuring spike alone may overestimate overall intensity of SARS-CoV-2 antibody signal by measuring spike alone given the increased availability of antigenic sites on the full spike ectodomain and possible cross-reactivity (Supplementary Fig. 9b). Including RBD in the protocol offers increased specificity and reduces the risk of false positives, whereas including spike offers increased sensitivity able to capture a polyclonal response. Therefore, using these antigens

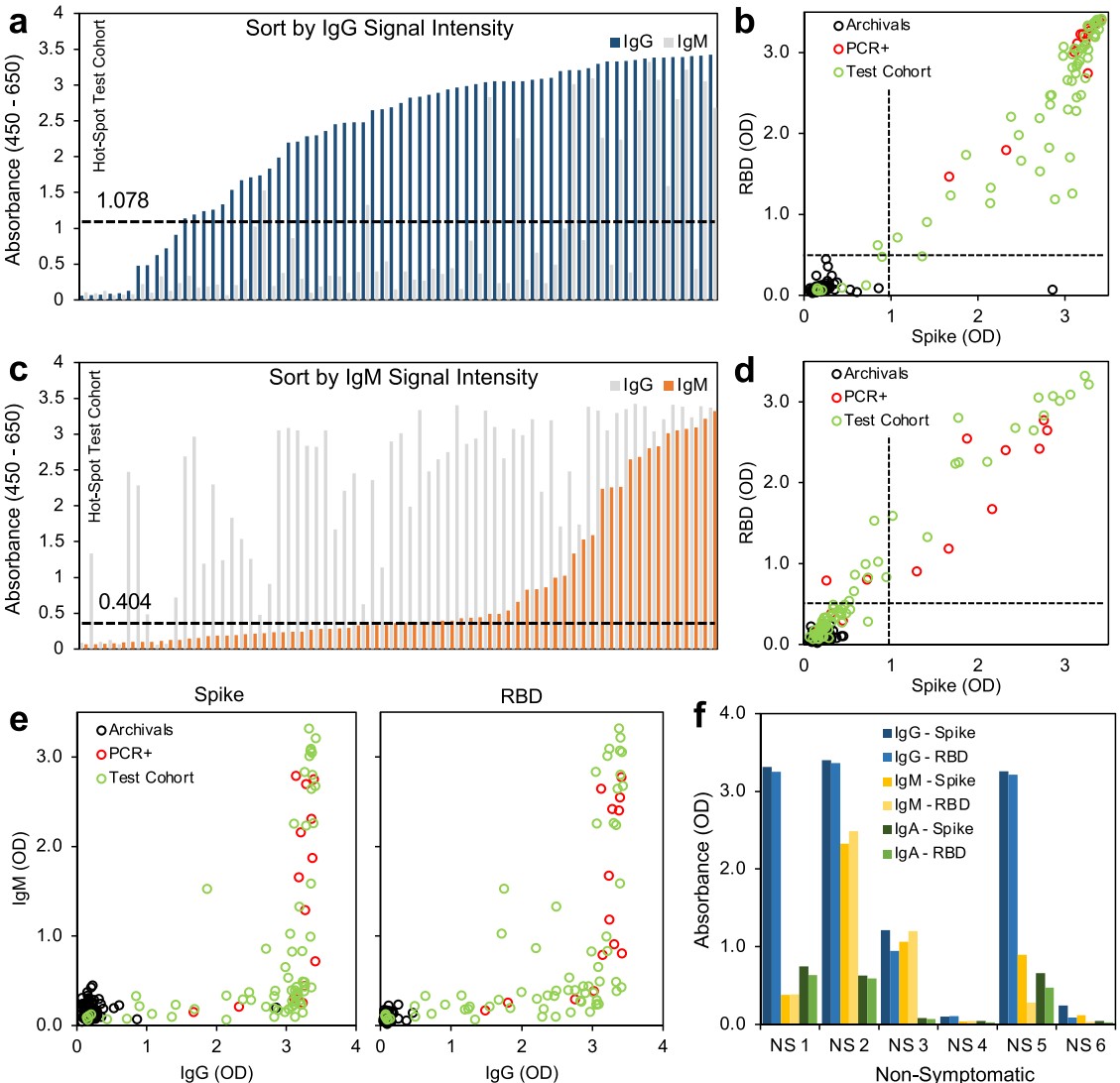

**Fig. 4 Small-scale testing of SARS-CoV-2 seropositivity from a hard-hit community. a** Signal intensity (absorbance) sorted by IgG (blue). **b** Thresholding of seropositivity in small-scale test cohort for SARS-CoV-2 IgG, black = archival negative controls, red = known PCR-diagnosed positive controls, green = test cohort. **c** Signal intensity (absorbance) sorted by IgM (orange). **d** Thresholding of seropositivity in small-scale test cohort for SARS-CoV-2 IgM. **e** Relation between IgM expression and IgG expression of spike and RBD antigens. **f** Seropositivity in non-symptomatic individuals who have not tested (PCR) positive for SARS-CoV-2 infection show robust IgG expression in absence of symptoms. $n = 68$ symptomatic donors, $n = 6$ non-symptomatic donors, IgG = blue, IgM = yellow, IgA = green. Source data are provided as a Source Data file.

individually risks false results, whereas together they provide the desired sensitivity and high specificity. Overall, we identified 37 potentially undiagnosed COVID19 cases in this sample data set (spike IgG+, RBD IgG+). Furthermore, as asymptomatic transmission has been reported[22], we tested six additional samples that were sourced from donors from this high-exposure community who displayed no symptoms, and were able to detect antibody in four of the six samples tested (Fig. 4f).

In order to further analyze this population and evaluate the potential for quantification, we developed standard curves of recombinant antibodies spiked into control seronegative blood and applied the resulting standard curve to quantify IgG and IgM concentrations (Fig. 5). Recombinant anti-RBD antibody was spiked into whole blood from two seronegative donors and either directly loaded onto microsamplers or spun down to isolate the serum from spiked blood (Fig. 5a). The resulting Spike and RBD ELISAs were in agreement between serum and microsamplers with a 1:1 signal ratio as seen previously, suggesting that microsamplers are a valid method of quantifying antibody in

blood (Fig. 5b). A sigmoidal four-parameter logistic curve was fit to the resulting optical density (OD) values to yield a standard curve of antibody concentration versus OD (Fig. 5c). Utilizing the resulting equations, we transformed the absorbance values of our small-scale test sample set to concentrations. The lower limits of detection/quantification for our assays were 0.3006 µg/ml (Spike IgG), 0.0656 µg/ml (RBD IgG), 0.1468 µg/ml (Spike IgM), and 0.1609 µg/ml (RBD IgM), which were equal or superior to prior published ELISA-based methods[23]. The upper limits of quantification at a dilution of 1:400 serum or 1:10 microsampler eluate into the ELISA were: 159.68 µg/ml (Spike IgG), 92.52 µg/ml (RBD IgG), 574 µg/ml (Spike IgM), and 116.28 µg/ml (RBD IgM; Fig. 5d, Supplementary Fig. 10). Above these concentrations, the detectors on the plate reader are saturated, and samples would need to be titered down for exact concentration measurement.

**Cross-reactivity of SARS-CoV-2 antibodies with other beta-coronaviruses and respiratory viruses.** To evaluate the potential for cross-reactivity that would alter the ELISA results, all control

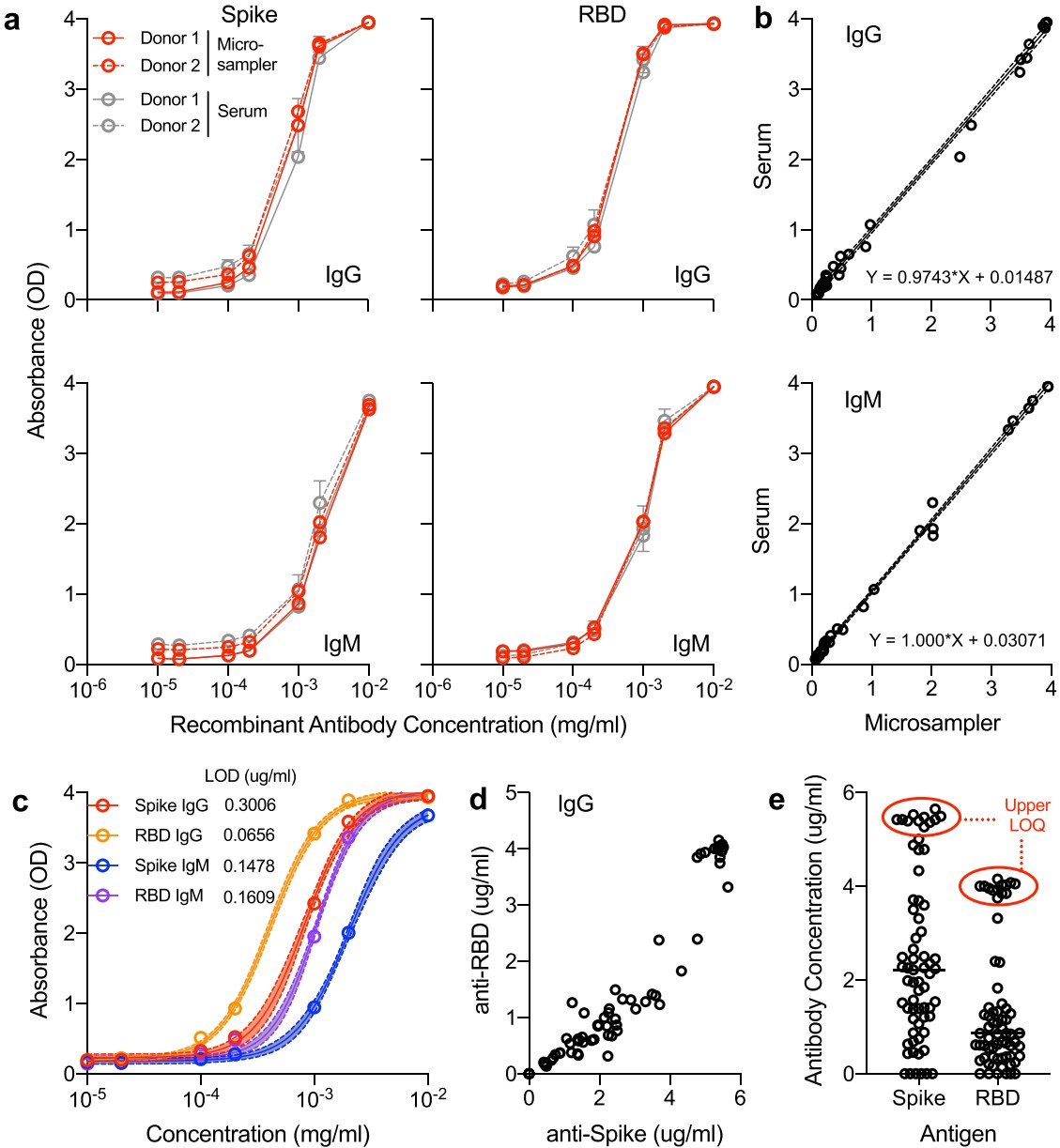

**Fig. 5 Quantification of antibody concentration utilizing 4PL sigmoidal model of recombinant antibody spiked into seronegative blood. a** Anti-RBD recombinant human antibody (IgG and IgM) was added to whole blood from two seronegative donors, then absorbed to microsamplers and remaining blood was spun down to isolate serum and analyzed on full spike ectodomain trimer (spike) or receptor-binding domain (RBD) ELISA. Data are mean ± SEM, $n = 3$, red = microsampler eluate, gray = matched serum. **b** Direct comparison of absorbance of range of recombinant antibody concentration in serum ($y$ axis) versus microsampler ($x$ axis) blood samples. **c** Sigmoidal four-parameter logistic (4PL) curve fitting to recombinant antibody dilution series, 95% confidence intervals shown shaded around fit curve, $n = 12$ replicates per data point, red = Spike IgG, orange = RBD IgG, blue = Spike IgM, purple = RBD IgM. **d** Quantification of IgG levels in a sample high-incidence population. **e** Upper limit of quantification at 1:400 dilution of serum into ELISA (1:10 dilution of microsampler eluate), $n = 68$. Source data are provided as a Source Data file.

samples were tested for the presence of antibody against the spike antigens of OC43, HKU1, MERS, and SARS1. We also tested all of the samples from our test sample set for these antibodies as well. When evaluating archival negative controls, we had previously shown very low signal for SARS-CoV-2 spike ELISA (negative, Fig. 3a). When compared with spike proteins from other coronaviruses, we detected very high OC43 and HKU1 antibodies across the pre-2019 sample set, concluding that there is minimal cross-reactivity between OC43 and HKU1 antibodies and the SARS-CoV-2 spike protein. Both OC43 and HKU1 spike antigens resulted in high absorbance readings in the test sample set ($3.07 \pm 0.22$ and $2.49 \pm 0.62$), SARS-CoV-2 positive controls

($3.25 \pm 0.16$ and $2.92 \pm 0.49$), and SARS-CoV-2 negative controls ($2.93 \pm 0.47$ and $2.30 \pm 0.65$), suggesting a strong prevalence of seasonal coronavirus infection (Fig. 6a, Supplementary Fig. 11). SARS1 and MERS spike antigen ELISAs resulted in significantly lower signal intensity when compared to SARS-CoV-2 (SARS-CoV-2 = $2.58 \pm 1.00$, SARS1 = $0.93 \pm 0.86$, MERS = $1.09 \pm 0.89$). However, as both proteins did induce higher than background levels of absorbance, we evaluated their expression on the negative controls. Overall minimal correlation ($R^2 = 0.3577$, $0.2606$, $0.2747$, and $0.4174$, correlation = $0.5981$, $0.5105$, $0.5241$, $0.6461$, respectively) was found between SARS-CoV-2 signal intensity and the signal intensity of the four other coronaviruses (OC43,

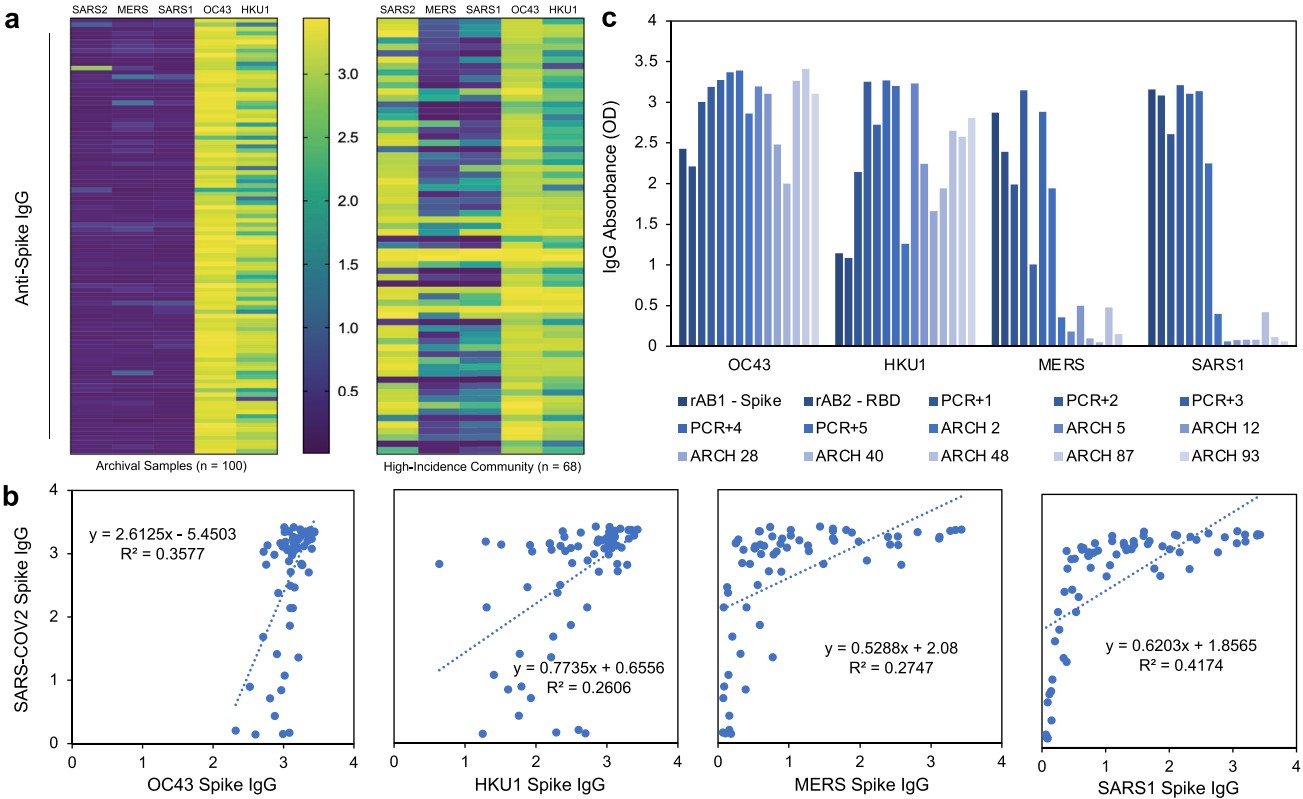

**Fig. 6 Cross-evaluation of high seroprevalence community samples with spike antigens from other coronaviruses. a** IgG absorbance of spike antigens from SARS-CoV-2, previous epidemic (MERS and SARS1), and seasonal (OC43 and HKU1) coronaviruses in archival pre-2019 samples (left) and high-incidence community (right). Color scale: yellow = high OD (4), purple = low OD (<0.5). **b** Scatter plots (with linear models and R-squared values) between IgG signal from SARS-CoV-2 spike and other coronavirus spike proteins. **c** Signal intensity for recombinant antibody control (rAB1 = spike monoclonal; rAB2 = RBD monoclonal), known SARS-CoV-2 nasal swab positive patient control (PCR+), and pre-2019 archival control (ARCH) samples for OC43, HKU1, MERS, and SARS1. Source data are provided as a Source Data file.

HKU1, MERS, SARS1). We have researched this phenomenon in depth and found that while minimal linear correlative effect is seen, there is potential reactivity between antibodies against SARS-CoV-2 spike protein and other spike proteins, and individuals who express high levels of these cross-reactive antibodies may be of interest to designers of pan-coronavirus therapeutics or vaccines[24]. Beyond coronaviruses, we also found no correlation between the hemagglutinin inhibition titer to H1N1 influenza A and the SARS-CoV-2 signal intensity, suggesting negligible cross-reactivity between antibodies against other common respiratory viruses and the antigens used in our assays (Supplementary Fig. 12). Overall, these data suggest minimal effect of cross-reactivity on the results of our SARS-CoV-2 serologic assays.

**Statistical simulation of confidence intervals in relation to validation of assays.** To evaluate our assay in the context of future serosurvey efforts, we modeled the statistical confidence over a range of disease prevalence and assay specificity. We simulated a study of 10,000 subjects and two strata and included options—based on varying scenarios—to alter stratum-specific prevalence, true sensitivity, true specificity, and negative control and positive control sample sizes. In all simulations, the confidence intervals included the true value over 95% of the replications. Figure 7 shows the plot of the lower and upper 95% confidence interval for the prevalence for each of the 1000 replications for each scenario. The supplemental materials provide the equations used to estimate the prevalence and 95% confidence intervals adjusted for weighting and estimated sensitivity and specificity. The confidence intervals are sorted by the

lower confidence interval. The figure shows that true specificity of 100% dramatically improves the width of the confidence intervals for all true prevalence rates. The figures also show an improvement as the negative control sample sizes increase from 100 to 300 and then 1000. This improvement is especially important in a low true prevalence scenario. The simulations with prevalence 0.001 (to simulate a true prevalence of near 0) shows that the lower bound will be 0 with a sample size of at least 300 negative controls to estimate specificity, and most upper bounds will be <1%. The simulations with prevalence of 1% again show the importance of a large sample size to estimate specificity. For negative control sample sizes of 1000 and a specificity of 1, the lower estimate is above 0 and the upper is almost always <1.5%. This will provide a very tight confidence interval in estimating the prevalence. These simulations show the importance of having high specificity and, if the expected prevalence rates are low, the importance of basing specificity estimates on large negative control sample sizes.

We further analyzed the impact of different sensitivity values and different sample sizes to estimate the sensitivity of the width of the 95% confidence intervals to the estimate of prevalence (Supplementary Fig. 13). This figure shows that there is minimal effect on the intervals for increasing positive control sample size above 100 and increasing sensitivity above 90%.

## Discussion

Understanding the seroprevalence of SARS-CoV-2 antibodies in the general population is critical to understand the extent of the spread of SARS-CoV-2 infection. However, the design of assays

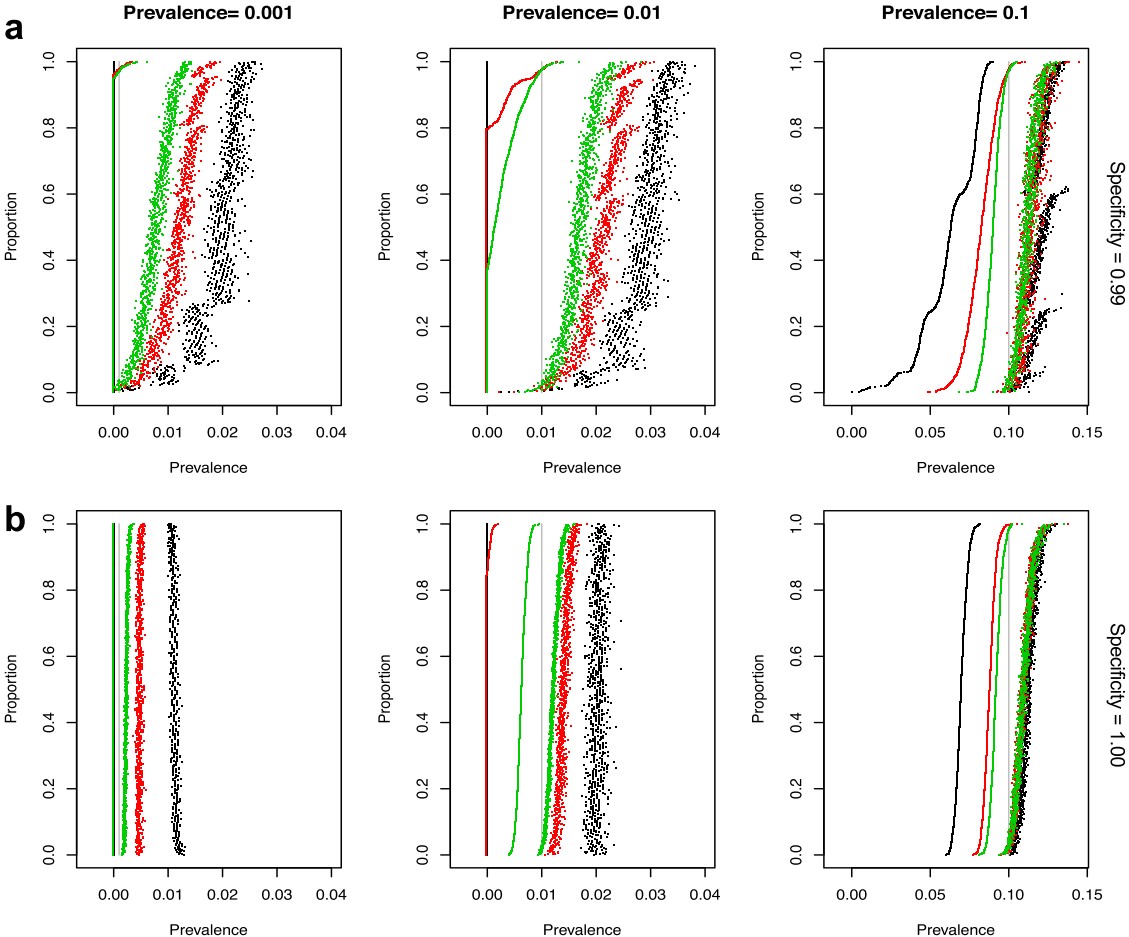

**Fig. 7 Simulation results showing confidence intervals for serosurvey prevalence calculations.** Each graph displays 95% confidence intervals (CI) for the estimate of prevalence from 1000 replications of each condition (including estimating the sensitivity and specificity for each replicate). The CIs are sorted by the lower bound, with lower bounds less than 0 replaced by zero. For all graphs, the true sensitivity is 0.90 and the simulations use 100 samples to estimate the sensitivity. For the graphs in row **a** the true specificity is 0.99 and in row **b** the true specificity is 1.00. The graphs show the simulations results with estimates of specificity using sample sizes of 100 (black), 300 (red), 1000 (green). Points are plotted black, red, then green, so some of the black and green points may be covered (e.g., the lower bounds for all three colors are all zeros in the bottom left panel). The columns give results for true prevalence values of 0.001, 0.01, or 0.1. The true prevalence for each simulation is shown by a vertical gray line.

and evaluation of seroprevalence data are not trivial. Owing to the variable nature of the human species, the technical intricacies of assay development, the potential impact of seroprevalence measurements on policy adjustment, and the current uncertainty of the correlation between antibody presence and immunity, these assays and trials must be approached with caution and rigor[25].

In this study, we compared the activity of four different constructs used in ELISA for serology of SARS-CoV-2 and established the ELISA assay being utilized in our seroprevalence study. First, we developed optimized protein expression and purification methods that could reproducibly generate spike and RBD proteins of consistent yield and quality. Although RBD production is fairly straightforward, the production of properly folded trimeric spike proteins is considerably more challenging. Spike proteins, likely owing to their heavy glycosylation, have a tendency to adsorb to membranes as well as purification media, requiring more complex methods to ensure consistent yield and quality. After experiencing significant protein loss during SEC using multiple purification resins, we decided to forgo this step and alter the preceding IMAC chromatography with a more complex elution scheme that minimized levels of contaminants and retained protein purity. This modified procedure has proved more consistent and allowed us to generate single large batches

(>40 mg) of spike protein from eight liters of cell culture media permitting tens of thousands of assays to be run on a single qualified batch of protein. Attention to quality of protein production and purification is critical for running a validated and specific ELISA, as contaminants in impure protein preparations or unstable proteins can yield decreased specificity and sensitivity in the assay.

Through careful evaluation of various ELISA assays and statistical determination of optimal threshold cutoffs for specificity, we determined that a combination approach using two ELISA assays, one employing the McLellan/VRC spike construct and the other employing the Ragon RBD, provided optimal results (Fig. 6). To be scored as "positive", both the spike IgG and RBD IgG OD levels or both the spike IgM and RBD IgM levels must be above their respective thresholds. Based on the data presented here, both the manual and semi-automated methods using our protocol provide a sensitivity and specificity of 100% when used both for IgG and IgM. As we deploy this method for a large-scale NIH serosurvey study, we will continue to test negative and positive control cases using the sampling method (microsamplers) and platform (semi-automated) planned for the study. As further control samples are tested, the thresholds can be further updated to ensure that the 95% lower confidence limit on

specificity is >99%. Final thresholds can then be determined and applied in future studies using those methods. The same methods could be applied by other groups using the sampling and ELISA methods of their choice, provided that the appropriate number of positive and negative controls are tested to determine the thresholds of positivity. In addition to the adjustment of these thresholds, estimates of prevalence from serosurveys using this method may use weighting methods (e.g., propensity weighting) and should adjust for the specificity and sensitivity estimates and variability about these estimates. The supplemental materials provide the equations that can be used to adjust the observed prevalence estimates based on the number of control samples tested.

Although we do not use the IgA to define seropositivity using this method, this addition would allow research to be performed (during serosurveys, prospective, or larger-scale testing) to better understand the development of immunity and the timing of various antibody responses[26]. The absorbances (OD) of IgA single-positive samples are very close to the threshold level, and no IgA single-positive donors were detected with an OD greater than 1. To ensure strong specificity we utilized IgG and IgM as the major antibody classes in the blood serum and utilized IgA to characterize the immune response.

The analysis of data from the small sample set collected from communities with high transmission rates in New York City and New Jersey demonstrated how people in various stages of their antibody responses may appear in our assays. There were 68 donors known to have exposure to and symptoms of COVID19, 22 of whom had also tested positive in the recent past for SARS-CoV-2 infection by PCR. Using our protocol, we identified 86% of these symptomatic and highly exposed individuals who were seropositive for IgG. We also identified 31 who were IgM positive while also being IgG positive (suggesting they were in an early stage of recovery from disease), whereas others were solely IgG-positive with IgM below the positive thresholds for spike and RBD (suggesting a later stage of convalescence)[27]. The overall expression of RBD and spike correlated well for both IgG and IgM, but several mid-range donors displayed lower RBD absorbance levels (still within positive range) when compared to spike, suggesting that these individuals may have a polyclonal antibody response that is not captured by the RBD antigen alone. Results from the utilization of dried blood correlated well with corresponding serum samples and support the use of dried blood techniques similar to those used widely in serology studies[28–31].

We have further shown, utilizing recombinant antibodies, that it is possible to derive a quantitation that is translatable across multiple assays; wherein a laboratory can run the same off-the-shelf commercially available recombinant controls on their assay of choice, and utilizing sigmoidal four-parameter logistic modeling, translate those OD readings into weight/volume measurements that can be compared from assay-to-assay. It is important to note that wing to the polyclonality and variability of human immune responses, along with antibody-to-antibody differences in binding affinity, these weight/volume concentration measurements may not be absolute but provide a benchmark for comparison across the diverse range of serologic tests currently utilized.

It is important to note that we developed this protocol with ease of adaptation by other labs in mind, and we focused on utilizing readily available reagents and instruments so that it could be applied easily in various resource settings. However, appropriate validation must be performed at each lab that adopts this protocol, owing to variances in equipment and reagents. These validations include determining the proper dilutions and building confidence intervals with positive and negative sample controls. This protocol, if validated and applied properly, can

provide sensitive and highly specific data that are more reliable than those of binary threshold assays and can serve to develop a more complete understanding of humoral immunity. This protocol is being implemented in our current NIH serosurvey, and we believe that it could offer a consistent method for others performing similar studies or expanded clinical antibody testing in the future.

## Methods

**Cloning and DNA preparation.** DNA for the expression of McLellan/VRC spike (VRC-SARS-CoV-2 S-2P-3C-His8-Strep2x2) and spike proteins for SARS-CoV, MERS-CoV, OC43-CoV, and HKU1-CoV were generously provided by Dr. Kizzmekia Corbett and Dr. Barney Graham (VRC, NIAID). DNA for the expression of Mt. Sinai spike (Kram-SARS-CoV-2 S-2P-His6) and Mt. Sinai RBD (Kram-SARS-CoV-2 S-RBD (319-541)-His6) were generously provided by Dr. Florian Krammer (Mt. Sinai School of Medicine) through BEI Resources. DNA for the expression of Ragon RBD (Ragon-SARS-CoV-2 S-RBD(319-529)-3C-His8-SBP) was generously provided by Dr. Aaron Schmidt (Ragon Institute of MGH, MIT, and Harvard[32]). Schematics of the similarities and differences in these constructs are shown in Fig. 2a. Transfection-quality DNA was produced in-house using the Qiagen Plasmid Plus Maxi Kit per the manufacturer's protocols or was generated at large-scale by Aldevron (Fargo, ND).

**Protein expression.** Manufacturer's protocols were followed for the transfection and culturing of Expi293 cells (ThermoFisher Scientific, Waltham MA). After 96 hours (spike proteins) or 72 hours (RBD proteins) of expression, culture supernatants (eight liters for spike proteins and four liters for RBD proteins) were clarified by centrifugation (4000 × g, 20 min, 4 °C) followed by filtration (Catalog# 12993, Pall Corporation, Port Washington, NY). Clarified supernatants were concentrated and buffer exchanged with the appropriate buffer by tangential flow filtration (TFF). Specifically, a MasterFlex peristaltic pump (Vernon Hills, IL) fed the clarified supernatant to either a 30 kDa MWCO cassette (Catalog# CDUF002LT, MilliporeSigma, Burlington, MA) for spike protein or 10 kDa MWCO cassette (Catalog# SK1P003W4, MilliporeSigma, Burlington, MA) for RBD. The clarified supernatant was concentrated to 10% of the initial volume and then buffer exchanged with 5 volumes of 1× PBS, pH 7.4 (Buffer A). Once the material was concentrated and buffer exchanged, the TFF cassette was rinsed with the appropriate buffer to collect any protein remaining in the cassette.

**Protein purification.** All chromatography was conducted at room temperature (~22 °C) using NGC medium-pressure chromatography systems from BioRad Laboratories Inc. (Hercules, CA). All spike proteins were purified similarly using IMAC followed by desalting into final buffer. Specifically, TFF-treated culture supernatant was adjusted to 25 mM imidazole and applied to a 5 ml Ni Sepharose high-performance nickel-charged column (GE Healthcare, Chicago, IL), previously equilibrated in Buffer A amended to 25 mM imidazole. The flow rate for all steps of the IMAC was 5 ml/min. After the load, the column was washed in Buffer A + 25 mM imidazole for three column volumes (CVs) and the protein eluted from the column by first reversing the orientation of the column, then by applying increasing concentrations of imidazole (beginning with 75 mM and increasing each step by 50 mM ending with 475 mM) in Buffer A. Elution fractions were analyzed by sodium dodecyl sulfate polyacrylamide gel electrophoresis (SDS-PAGE) and Coomassie-staining and appropriate fractions were pooled. The sample was desalted into the final buffer of 1× PBS, pH 7.4, (10× PBS 70011069 ThermoFisher Scientific, Waltham, MA) using a HiPrep 53 ml 26/10 desalting column (GE Healthcare, Chicago, IL) with 14 ml injections at 9 ml/min for all steps. The final protein sample was created by combining the bulk elutions from multiple runs of the desalting column. The protein concentration was determined by measuring the $A_{280}$ using a Nanodrop One spectrophotometer (ThermoFisher Scientific, MA, USA). Final protein was dispensed as 0.5 ml and 0.05 ml aliquots, snap frozen in liquid nitrogen, and stored at −80 °C. One ml of the final protein was thawed and analyzed by analytical SEC with a 10/300 Superdex200 analytical column (GE Healthcare, Chicago, IL), using a flow rate of 0.5 ml/min, to confirm the trimeric nature of the protein. Transmission electron microscopy of the purified McLellan/VRC and Mt. Sinai spike proteins was also carried out by dilution of the samples to 0.02 mg/mL in 20 mM Tris-HCl, pH 8.0, 200 mM NaCl followed by loading onto glow-discharged carbon film grids. Grids were washed twice in buffer, stained with 0.75% w/v uranyl formate (pH 4.5) four times, and stain was then wicked off-the grid and grids were dried under an incandescent lamp. Stained grids were imaged on a Hitachi 7650 electron microscope at ×40,000 magnification.

Mt. Sinai RBD and Ragon RBD proteins were purified similarly using IMAC followed by SEC into final buffer using NGC medium-pressure chromatography systems from BioRad Laboratories Inc. (Hercules, CA). Specifically, TFF-treated culture supernatant was adjusted to 25 mM imidazole and applied to a 10 ml Ni Sepharose high-performance nickel-charged column (GE Healthcare, Chicago, IL), previously equilibrated in 1× PBS, pH 7.4, amended to 25 mM imidazole. The flow rate for all steps of the IMAC was 5 ml/min. The column was washed in 1× PBS, pH 7.4, 25 mM imidazole for four CVs. Proteins were eluted from the column with

a gradient of 1× PBS, pH 7.4, from 25 mM to 500 mM imidazole over 20 CVs. Elution fractions were analyzed by SDS-PAGE and Coomassie-staining and appropriate fractions were pooled. The sample (~40–50 ml) was concentrated to ~5 ml using 10 kDa MWCO Amicon Ultra centrifugation filter units (MilliporeSigma, Burlington, MA), and the sample applied to a 16/60 Superdex75 SEC column (GE Healthcare, Chicago, IL) equilibrated in final buffer of 1× PBS, pH 7.4. The column was developed at 1 ml/min and one ml fractions were collected. Fractions were analyzed by SDS-PAGE and Coomassie-staining and appropriate fractions were pooled and filtered with a 0.22 µM syringe filter (low protein binding). The protein concentration was determined by measuring $A_{280}$ using a Nanodrop One spectrophotometer (ThermoFisher Scientific, MA, USA). Final protein was dispensed as 0.25 ml and 0.05 ml aliquots, snap frozen in liquid nitrogen, and stored at −80 °C. Protein sequences are available in Supplemental Data 1.

### Enzyme-linked immunosorbent assay.
Serum samples collected via venipuncture in serum separator tubes were processed and stored in 200ul aliquots at −80 °C before analysis. Whole blood samples were loaded onto 20 µl NeoteryX Mitra Microsampling device tips, dried completely, and transferred into 500 µl Eppendorf tubes that were stored dry at −80 °C until elution. All negative control blood samples were used under a clinical protocol (NCT01386424) approved by the National Institute of Allergy and Infectious Diseases Institutional Review Board (IRB) and conducted in accordance with the provisions of the Declaration of Helsinki and Good Clinical Practice guidelines. All participants signed written informed consent prior to enrollment. All other samples were collected under an IRB exemption since these were fully deidentified samples. The results of this study are not a part of the referenced national serologic survey NCT04334954.

Prior to analysis via ELISA, one microsampler tip (20 µl) was added to 400 µl of 1× PBS (Gibco) + 1.0% BSA (Sigma) + 0.5% Tween20 (Sigma) in a 1 ml deep-well 96-well plate (ThermoFisher). The plate was then covered with an adhesive foil seal and incubated overnight at 4 °C on a shaker at 300 rpm. The resulting eluate was used immediately for ELISA or stored at −80 °C until use.

One hundred (100) microliters of spike (1 µg/ml) or RBD (2 µg/ml) antigen was added into 96-well Nunc MaxiSorp ELISA plates (ThermoFisher) and incubated for 16 hours at 4 °C. Wells were washed with 300 µl of 1× PBS + 0.05% Tween20 (PBST) three times and blocked with 200 µL PBST + 5.0% Non-fat Dry Milk (blocking buffer) for 2 hours at room temperature. The plate was then washed three times with 300 µl PBST. One hundred microliters of each sample were added in technical duplicate (serum diluted 1:400 in blocking buffer, microsampler eluate diluted 1:10 in 1× PBS + 5.0% Non-Fat Dry Milk) to the plate. Controls on each plate included blank/secondary only control, known nasal swab positive serum control, archival serum negative control, and recombinant antibody positive controls (NIAID VRC). Samples were incubated at room temperature for 1 hour, and plates were washed three times with 300 µl PBST. Secondary antibody (horseradish peroxidase (HRP) conjugated: goat anti-Human IgG (H+L) cross-adsorbed secondary antibody, goat anti-Human IgM cross-adsorbed secondary antibody, goat anti-Human IgA Cross-Adsorbed Secondary Antibody; ThermoFisher) was diluted at 1:4000 in blocking buffer and 100 µl of each antibody was then added to each well and incubated for 1 hour. Plates were washed three times with PBST, then incubated with 1-Step™ Ultra TMB-ELISA Substrate Solution (ThermoFisher) for 10 min prior to stopping the reaction with 1 N sulfuric acid Stop Solution (ThermoFisher). Plates were then read for absorbance at 450 and 650 nm (PHERAstar FSX plate reader) within 30 min of stopping the reaction. Absorbance (OD) is calculated as the absorbance at 450 nm minus the absorbance at 650 nm to remove background prior to statistical analysis.

### Defining thresholds for positivity.
To evaluate the specificity of ELISA assays and establish thresholds for positivity, serum samples collected from well-characterized healthy volunteers in NIH study NCT01386424 prior to 2019 were obtained as negative controls for SARS-CoV-2 to define the threshold for seropositivity and evaluate specificity ($n = 100$ for manual ELISA, $n = 300$ for semi-automated ELISA). Convalescent blood samples from known SARS-CoV-2 nasal swab PCR+ donors were obtained to evaluate sensitivity ($n = 14$ for manual ELISA, $n = 46$ for semi-automated ELISA). A test sample set for use to further assess the overall serologic assays testing protocol was also obtained. This set of 74 deidentified blood samples was acquired from a blood drive in very high-risk Jewish communities in New York and New Jersey. There were 22 donors who reported a previous PCR+ diagnosis of COVID19, 46 who reported a high chance of exposure as well as recent symptoms, and six donors reporting no recent symptoms but a high chance of exposure. These samples were tested for the presence of SARS-CoV-2 antibody and were also run against a panel of spike proteins from four other beta-coronaviruses; MERS, SARS1, OC43, and HKU1 to establish seroprevalence estimates for these pre-pandemic coronaviruses in a current population. This population was a test cohort for our assay, and it was unknown if each individual donor was seropositive or previously infected by SARS-CoV-2 based on a PCR-based detection strategy.

### Quantification of OD values via standard curve of recombinant antibody.
Healthy seronegative whole blood collected in EDTA-coated tubes was spiked with commercially available human chimeric anti-RBD IgG and IgM antibodies (GenScript, Cat# A02038, A02046). One (1) mg/ml stock antibody was diluted at 1:100000, 1:50000, 1:10000, 1:5000, 1:1000, 1:500, and 1:100 in whole blood. The resulting spiked blood was loaded onto Neoteryx 20 µl Microsampling devices and allowed to dry overnight at room temperature. The remaining blood was then processed and spun down at $1000 \times g$ for 10 min to isolate the serum. Microsamplers were eluted as per previously described and both serum and eluate were run on ELISA as per aforementioned protocols. Resulting data were plotted as concentration (x axis) versus OD (OD, y axis) in GraphPad Prism (v 9.0.0), then fit to a sigmoidal 4 parameter logistic (4PL) model. The sigmoidal 4PL equations were then used to calculate the concentration of example convalescent samples.

### Semi-automation of ELISA.
The BioTek Instruments EL406 washer/dispenser used is equipped with various liquid handling capabilities, including two syringe pump dispense heads with 16 nozzles each, a peristaltic pump dispense option that can be outfitted with an eight-tip bulk dispense cassette and, a 96-channel wash head for aspirating and dispensing. The unit is outfitted with a 30-plate stacker to input plates, returning them to the output stack when finished with a protocol step as well as a restack function to return the plates to the original order. There are four 4 L bottles that can be used to dispense from and clean the 96-channel wash head using a valve module to switch between the different source bottles. Two waste collection vessels, a primary as well as a secondary containing a liquid level sensor to alert when emptying is necessary, are also plumbed to the washer/dispenser. The unit used for these experiments has an integrated BioStack 4 microplate stacker, but this is not necessary to perform the described protocol; loading the plates manually onto the BioTek EL406 plate nest will yield the same liquid handling results. It is important to note that the aspiration heights described within this process were optimized for the particular 96-well plate used in this ELISA protocol and would need to be re-optimized if a different assay plate were used.

This process will only describe semi-automated steps and will not detail manual segments. (1) 96-well plates are currently coated manually with the Spike or RBD protein using a multichannel pipette, but alternatively the peristaltic eight-tip small volume dispense cassette on the BioTek EL406 will be incorporated in the future. (2) After 16 hours of incubation at 4 °C the plates are loaded into the input stack of the washer/dispenser, the syringe B dispense line is primed with blocking solution, and the 96-channel wash head is primed with PBST wash solution. Prior to each step, priming the associated dispense lines and wash head must occur. (3+4) The blocking protocol is run for the loaded plates, which entails washing all wells three times (3×) with PBST followed by dispensing 200 µL of blocking solution. In subsequent steps, "washing" refers specifically to aspirating the existing liquid, in this case coating protein, from the well and dispensing 300 µL of PBST into the same wells using the 96-channel wash head on the BioTek EL406 washer/dispenser. The height of the aspiration pins for the first two wash cycles is 3.68 mm above the BioTek plate carrier and the last aspiration happens slightly lower at a height of 3.43 mm above the plate carrier with an additional secondary aspirate at the same height 2.06 mm to the left of the well center in order to remove the remaining liquid on the well perimeter prior to blocking solution addition. For reference, the well bottom is ~3.4 mm above the plate carrier. Once all plates have been washed with block added into the wells and returned to the output stack, all plates are then restacked into the original order into the input stack.

It is necessary to clean the syringe B dispense head and line soon after the block has been added to all plates to prevent clogging of the dispense tips. Each time a dispense happens throughout this protocol it will be assumed that the following cleaning protocol has been performed for the syringe head used. The bottle cap with the liquid sourcing end of line is fastened into an empty bottle and primed with 8 ml of air at atmospheric pressure two times sequentially; next the same cap and line are secured onto a bottle containing deionized water and primed with 8 ml of the water five times, a clean empty bottle is then hooked up again and primed with 8 ml of air two times; next the cap and line are placed onto a bottle containing 70% ethanol and primed five consecutive times with 8 ml of the ethanol, followed again with two 8 ml primes of air and a final deionized water rinse priming 8 ml of water five times. Before a new reagent is primed through the respective line two primes of 8 ml of air using a clean empty bottle are completed for separation of water and new reagent.

(5) After the 2-hour incubation, (6) the plates are washed 3× with PBST to remove blocking solution and roughly 30 µL is left in all wells to prevent drying before sample addition. Here on out, it will be assumed that incubation steps happening at room temperature will be stacked in the BioTek input stacker with a lid, which will be removed before beginning next step, on the top plate. The height of the aspiration pins for the first two wash cycles is 3.68 mm above the carrier and the final aspiration happens at 4.70 mm above the plate carrier in order to retain some wash fluid, as mentioned. (7) Currently, samples have been added to the ELISA plates using a multichannel pipette; but for the scaled-up process an Integra Biosciences VIAFLO electronic 96-channel pipette can be used to transfer 100 µL of the samples from a 96 deep-well block into multiple ELISA destination plates.

(8) Once the 1-hour incubation ends (9+10) all wells in the plates within the associated batch will be washed 3× with PBST followed by the dispensing 100 µL of the respective antibody into all wells of the 96-well plate using the BioTek EL406 washer/dispenser. The height of the aspiration pins for all three wash cycles is 3.43 mm above the plate carrier. There is also a secondary aspiration at the

same height 2.06 mm to the left of well center. Each addition of PBST wash fluid also incorporates a 15 s soak prior to aspiration. This washing procedure is critical to decreasing background and fully removing the samples (primary antibody). Plates are restacked into the original order within the input stack.

(11) After the 1-hour incubation, (12) all wells in the batch of plates are washed 2× with PBST, and the final dispense of 300 μL is added using the 96-channel wash head. The height of the aspiration pins for these two cycles is 3.68 mm above the carrier. There is also a secondary aspiration at the same height 2.06 mm to the left of well center. Each addition of PBST wash fluid incorporates a 25 s soak prior to aspiration. The difference in soak time from the previous process of washing the samples out of the wells allows the timing to stay consistent for each assay plate. Plates with the remaining 300 μL of PBST are then restacked into the original order within the input stack.

Performing the initial two washes with PBST and leaving the last 300 μL in the wells enables the addition of substrate to happen quicker than if all three washes were to happen prior to substrate addition. With the critical nature of the 10 min incubation for substrate and to increase throughput, it is desired to systematically increase the speed of substrate addition as in this protocol.

(13) The remaining PBST is removed from all wells of the plates using the BioTek 96-channel wash head. A height of 3.43 mm above the plate carrier is set for the aspiration pins, and a secondary aspirate at the same height 2.06 mm to the left of well center is performed. A volume of 100 μL of substrate is added to all wells using syringe A dispense head, which is the designated line used for substrate only, as it is a sensitive reagent. As the loaded plates are dispensed from the input stack, they are returned to the output stack, and after the last plate is finished all plates are restacked in the original order within the input stack. (14) Each plate is incubated with the substrate solution for 10 min.

(15) It is critical that the stop solution is added after the 10-minute incubation, therefore, the plate batch size at this step is adjusted accordingly. As soon as the first plate has finished incubating with substrate the BioTek protocol for adding 100 μL of the stop solution using syringe B dispense head is started. An important note regarding the stop solution used in this ELISA protocol is that, when 100 μL was set for the dispensing variable, ~60 μL was actually added to the wells. In order to compensate for the characteristics of this liquid 175 μL was set as the dispense variable for the stop solution addition. It will be checked regularly for volume accuracy. Once all plates have received stop solution, they are restacked into the original order into the input stack.

(16) Absorbance is measured at 450 nm and 650 nm. Data were initially collected using the BMG Labtech PHERAstar FSX, and the BioTek Epoch 2 has subsequently been used. The BioTek stacker containing the plates that have been dispensed with stop solution and returned to the original order is removed from the BioTek input stack position and placed into the input stack position for the Epoch 2. The read protocol is started, and plates are fed into the Epoch 2 for data collection. The BioTek EL406 96-channel wash head is cleaned at the end of each day to ensure no reagent buildup or clogging happens. This entails flushing 350 ml of deionized water followed by 350 ml of 70% ethanol and finally with 350 ml of deionized water. The 96-channel wash head dispense tips as well as the aspiration pins are cleaned in this process.

Based on thorough titering studies of both archival and PCR+ SARS-CoV-2-diagnosed patient sera (Supplementary Fig. 7), we determined that the optimal titrations for the serum sample into ELISA were 1:400. Microsampler dilution was adjusted accordingly to 1:10 into PBS + 5.0% NFDM, to ultimately result in 0.05% Tween20 in the final diluted sample.

**Statistical analysis and determination of sensitivity and specificity**. To determine confidence intervals for specificity and sensitivity, we used exact binomial methods, which (to give accurate coverage) require an independent data set to determine the threshold. Because we used the same data set to determine the threshold and evaluate sensitivity and specificity, we only considered a restricted class of thresholds, determined by adding an integer value of standard deviation to the mean for the negative controls. This approximates accurate confidence intervals. The manual and semi-automated ELISAs were evaluated using this procedure for SARS-CoV-2 spike and RBD for IgG and IgM to determine overall sensitivity/specificity of identifying a seropositive sample. For each application, further negative control samples beyond what is presented here can be evaluated to allow for a threshold to be chosen so that the lower 95% confidence limit of specificity is >99%.

To evaluate how well our final testing protocol would work in practice, simulation studies were performed to determine how the number of samples used to estimate sensitivity and specificity affect the confidence interval of the prevalence in a serosurvey. Several scenarios were simulated to evaluate the performance of the estimation methods with simple weighting and different sample sizes used to estimate the sensitivity and specificity. In the simulations, data were generated from a survey size of 10,000 under overall true prevalence rates of 0.1%, 1%, and 10%. Scenarios with two different sets of observed weights were evaluated, with samples drawn from two populations with the true proportion of people in each population being 0.35 and 0.65 but were drawn equally from each population. The weights for individuals in the smaller and larger populations were ~0.7 and 1.3, respectively. In the population with the smaller size the true prevalence rate was inflated above the

overall rate, while the true prevalence in the second population was below the overall rate. A second set of more extreme weights was examined. In this simulation the proportion in each population were 5% and 95%, which corresponded to weights of 0.1 and 1.9, respectively.

We performed simulations with true specificity values of 99% and 100%, using sample sizes of 100, 300, and 1000 to estimate the specificity. A true sensitivity of 90% was used in these simulations with a sample size of 100 used for the estimate. Another set of simulations was performed with true sensitivity values of 90R and 95% and sample sizes of 100, 200, and 300 used in the estimate with true specificity set to 99% with sample size of 1000 used in the estimate. We performed 1000 simulations for each scenario and calculated 95% confidence intervals around the adjusted prevalence estimate for each replication of the simulation.

**Statistical modeling**. Let $r_1$, $r_2$ be the values from the census for proportion of people in each population. Let $n$ be the number of people in the sample. Let $y_{1i}$ be the $i$th sampled response and $n_1$ the number of observations from population 1 and $y_{2i}$ be the $i$th sampled response and $n_2$ the number of observations from population 2. The weights for an observation in population 1 or 2 are $w_1 = \frac{r_1}{n_1/n}$ and $w_2 = \frac{r_2}{n_2/n}$ The unadjusted estimate of the prevalence is the weighted estimate of the mean (Equation 1), which is

$$\tilde{p} = \frac{\sum_{i=1}^{n_1} w_1 y_{1i} + \sum_{i=1}^{n_2} w_2 y_{2i}}{\sum_{i=1}^{2} \sum_{j=1}^{n_i} w_i} = \frac{n_1 w_1 \hat{p}_1 + n_2 w_2 \hat{p}_2}{n} \quad (1)$$

where $\hat{p}_i$ are the observed proportions in each population. The variance of $\tilde{p}$ (Equation (2) is

$$\widehat{var}(\tilde{p}) = \frac{n_1 w_1^2 \hat{p}_1 (1 - \hat{p}_1) + n_2 w_2^2 \hat{p}_2 (1 - \hat{p}_2)}{n^2} \quad (2)$$

Lang and Reiczigel[33] (2014) provide a prevalence estimate that adjusts for the sensitivity and specificity of the assay, taking into account the variability of the estimate of prevalence as well as the variability in the estimation of sensitivity and specificity. Thus, the Lang-Reiczigel method requires the sample sizes for validation of the assay, specifically, the number of true positives used to estimate the sensitivity, and the number of true negatives used to estimate the specificity. Although the method was developed for simple random samples, we make the following modification for other survey designs. Let $\tilde{p}$ and $\widehat{var}(\tilde{p})$ be the prevalence and variance estimates under the design with perfect sensitivity and specificity. We wish to the know the sample size for the estimate of $\tilde{p}$ that would equal the variance of $\widehat{var}(\tilde{p})$. Therefore, set

$$\widehat{var}(\tilde{p}) = \frac{\tilde{p}(1 - \tilde{p})}{n_{eff}},$$

where the righthand side of the equation is modeled after the binomial variance from a simple random sample. In other words, the effective sample size is the simple random sample size that would give similar variance to the variance of the weighted design. Then, we solve for $n_{eff}$ so

$$n_{eff} = \frac{\tilde{p}(1 - \tilde{p})}{\widehat{var}(\tilde{p})}$$

Then, we input $\tilde{p}$ and $n_{eff}$ into the Lang-Reiczigel method in place of the simple random sample prevalence estimate, $\hat{p}$, and its sample size, $n$.

Here are the Lang-Reiczigel equations with those modifications[1]. The estimate of prevalence adjusted for sensitivity and specificity is

$$\hat{\phi} = \frac{\tilde{p}' + S_p' - 1}{S_p' + S_p' - 1}$$

with $n'_{eff} = n_{eff} + z^2$, $\tilde{p}' = \frac{n_{eff}\hat{p} + z^2/2}{n'_{eff}}$, $n'_{Sp} = n_{Sp} + 2$, $Sp' = \frac{n_{Sp}Sp+1}{n_{Sp}+2}$, $Se' = \frac{n_{Se}Se+1}{n_{Se}+2}$ with $Sp$ and $Se$ being the estimates of specificity and sensitivity, and $n_{Sp}$ and $n_{Se}$ being the number of observations used to estimate specificity and sensitivity, with $z$ being the upper quantile from the standard normal distribution with probability $1 - \alpha/2$, where $\alpha = 1 - q$ and $q$ is the confidence level for the confidence interval (e.g., 95%). Then

$$\widehat{var}(prev) = \frac{\tilde{p}'(1 - \tilde{p}')/n'_{eff} + \hat{\phi}^2 * Se'(1 - Se')/n'_{Se} + (1 - \hat{\phi})^2 Sp'(1 - Sp')/n'_{Sp}}{(Se' + Sp' - 1)^2}.$$

The estimate of the confidence intervals is given by

$$\hat{\phi} + k \pm z \sqrt{\widehat{var}(prev)}$$

with

$$k = 2z^2 \left[ \frac{\hat{\phi} Se'(1 - Se')}{n'_{Se}} - \frac{(1 - \hat{\phi}) Sp'(1 - Sp')}{n'_{Sp}} \right].$$

**Reporting summary**. Further information on research design is available in the Nature Research Reporting Summary linked to this article.

## Data availability

The data that support the findings of this study are available from the corresponding author upon reasonable request. Source data are provided with this paper.

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

## Acknowledgements

The authors would like to acknowledge Dr. Kizzmekia Corbett and Dr. Barney Graham of the NIAID VRC for their generous donation of coronavirus spike plasmids and recombinant antibodies, and Dr. Aaron Schmidt, Jared Feldman, Blake M. Hauser, and Timothy M. Caradonna for their donation of their RBD expression plasmid. We would like to thank Golan Ben-Oni, Rabbi Shua Brook, Dr. Adam Polinger, Dr. Avi Rosenberg, and the Jewish community of New York and New Jersey for their generous donation of blood samples used to validate and test this assay. This research was supported in part by the Intramural Research Program of the NIH, including the National Institute of Biomedical Imaging and Bioengineering, the National Institute of Allergy and Infectious Diseases, and the National Center for Advancing Translational Sciences. This project has been funded in part with Federal funds from the National Cancer Institute, National Institutes of Health, under contract number HHSN261200800001E. Disclaimer: the NIH, its officers, and employees do not recommend or endorse any company, product, or service.

## Author contributions

C.K.T., H.K., D.E., M.J.M., M.D.H., and K.S. conceptualized and designed the study. C.K.T., H.K., M.D., K.S., J.M., A.S., V.W., P.F., J.P.D., M.H., G.G., S.M., J.H., and K.S. performed experiments. S.H. and M.P.F. performed statistical analyses. S.M., W.G., M.D.H., M.J.M., D.E., and K.S. provided scholarly input on study design and implementation. K.S. supervised the study.

## Competing interests

The authors declare no competing interests.
