## [Peer Review File · Nature Communications]

REVIEWER COMMENTS

Reviewer #1 (Viral immunity, antibody response.) (Remarks to the Author):

Standardization of enzyme-linked immunosorbent assays for serosurveys of the SARS-CoV-2 pandemic using clinical and at-home blood sampling

The authors have designed and described in detail a novel ELISA protocol for detecting SARS-COV-2 in this paper. The authors have developed a pipeline for producing and validating several SARS-COV-2 antigens from different sources for their ELISA protocol, which will be useful for any labs repeating their ELISA or producing their own ELISA. The authors demonstrated that their ELISA has a high specificity and high sensitivity when tested against their archived negative samples and convalescent blood samples from previously SARS-COV-2 PCR+ individuals. Additionally, the authors showed that their ELISA still works reasonably well for identifying SARS-COV-2 seropositive individuals with samples collected from a hard-hit community. To elucidate if seasonal coronavirus infections could affect their ELISA results, the authors tested OC43, HKU1, SARS-COV-1, and MERS spike proteins and confirmed there was low reactivity in the ELISA. Finally, the authors did a simulation analysis to illustrate how sample size and the various specificities of their ELISA could affect the confidence interval of seroprevalence study.

This manuscript provides a useful protein production and a simple ELISA protocol that can be performed manually or with the assistance of automation. An easy to conduct ELISA assay with high specificity and high sensitivity is useful, as early SARS-COV-2 serology studies have failed to have both specificity and sensitivity. Unfortunately, this ELISA protocol and paper is not per-se novel. ELISA assays are widely available now that have both characteristics- dampening enthusiasm. Moreover, there are a few issues which should be addressed. My critiques are listed below.

Comments:

- 1) The supplementary figures are out of order. Supplementary figure 1 is cited first in the results section, followed by Supplementary Figure 6. Please reorder the supplementary figures so they follow the text.
- 2) Was specificity and sensitivity calculated from micro sampler elute for Supplementary Figure 2? The data correlates but a few samples deviate in IgM. As this ELISA protocol may be used for seroprevalance studies, which could receive a wide variety of samples; a thorough characterization of a variety of sample inputs would be useful.
- 3) The authors state "Overall no correlative effect ($R^2 = 0.3577, 0.2606, 0.2747, \text{ and } 0.4174$, correlation = $0.5981, 0.5105, 0.5241, 0.6461$ respectively) was found between SARS-CoV-2 signal intensity and the of the four other coronaviruses (OC43, HKU1, MERS, SARS1)." This statement needs to be changed. No correlation would be $R^2=0$. There is a correlation between the signal intensity of SARS-COV2 and the other coronaviruses.
- 4) Figure 6 has several issues that should be addressed. The utility of the figure is questionable since the text nor the figure make an easily understandable argument for why this analysis demonstrates the usefulness of their specific ELISA in a seroprevalance study. Are the assumptions the simulation makes reasonable to assume for a large scale serology study conducted with this assay? Are 100, 300, or 1000 samples the normal sample size for their serology study? Is a 99% sensitivity expected during a large scale serology study when the samples come from multiple sources and from patients with a variety of symptoms and dates from infection/symptom onset? In addition, the labeling and description of the red/green/black lines, red/green/black dots, and proportion-axis label is not explained well. Finally, the third graph in Fig6a and b have the green and black lines closer together

than the red. This means the black line (100 samples) and green line (1000 samples) agree well but the red line (300 samples) does not, which seems incorrect.

5) The authors state "This suggests that not only could detection of antibodies be performed reliably, but quantification of antibodies from these mail-in sampling devices was possible (Supplementary Figure 2)." The authors never demonstrate quantifying any of their results. The authors should demonstrate that their assay has a sufficient dynamic range to quantify antibodies in samples. This would further help the paper as a protocol and reagents for quantifying could be included.

6) Out of the 68 high incidence community, 62 were found to be IgG positive. Out of the remaining 6, all were IgA positive in Supplementary Figure 11k. Should the authors change their strategy from analyzing IgM and IgG, to IgG and IgA as this seems to detect more positive samples, or is there a potential loss of specificity with IgA?

7) Supplementary Fig 7 needs better labels. Are the dots replicates and the x-axis the well positions, or vice versa. Also, were these archival samples or other samples that were tested? Was intermediate precision, day-to-day variability, operator-to-operator variability, laboratory-to-laboratory variability, assessed in addition to repeatability?

8) The authors state, "The adaptation of this protocol to analyze both IgM and IgA allows characterization of the stage of infection and increases the sensitivity to identify relatively early-stage infections that have yet to mount a strong IgG response". While it is true that IgM and IgA can be used to characterize the stage of infection in other infections, the authors have provided no data that their assay is sensitive enough to detect an early-stage of infection using IgM or IgA before an IgG response has been mounted. This statement should be removed.

9) The authors state "Based on the data presented here, both the manual and semi-automated methods using our protocol provide a sensitivity and specificity of 100% when used both for IgG and IgM." Their data with their archived samples and PCR+ controls have a sensitivity and specificity of 100%, but their hard-hit community data does not have a sensitivity of 100% using IgG and IgM. This statement should be modified to "based on our archived negatives and convalescent blood samples".

10) A large scale seroprevalence study will encounter many confounding variables such as: severity of infection, duration since infection, duration since symptom onset, age, ethnicity, gender, sample input, sample collection method, time since collection, concurrent non-COVID infections, preexisting conditions, individuals with autoimmune disorders, individuals on immunosuppressants, and more. The authors have not addressed how any of these confounding variables could affect their sensitivity or specificity. This means their specificity and sensitivity could be drastically altered when analyzing large scale sample sets, which could dramatically affect their final results.

Reviewer #2 (Antibody response, Ag mimicry, vaccine.) (Remarks to the Author):

The manuscript by Klumpp-Thomas et al entitled "Standardization of" describes a protocol an ELISA-based to determine thresholds of antibodies that defines seropositivity of serum samples of SARS-CoV-2 patients. The approach described summaries that being conducted by the national serosurvey (sponsored by the NIH) with the intent to standard large cohort studies.

Positive aspects of the manuscript

- 1.) A validated approach to determine immunity and correlates of protection towards and against the virus.
- 2.) Multiple sample types and approaches tested.
- 3.) Good statistical analyses of the results
- 4.) Results support the conclusion drawn.

5.) Results observed is similar either by manual or semi-automated procedures.

The manuscript certainly describes the ongoing approach taken in the national Immunosurvey cohort study and well describes the protocol taken. However, the expectation is that this protocol be adopted by multiple labs. Yet the antigens used to develop the protocol may not be readily available and still requires some skill to generate the two antigens that are important to determine the sensitivity described in the paper. Little discussion is directed toward this expectation. Over all the manuscript is well written and provides the technical ground work required for validated analyses of antibodies to the virus spike protein.

RESPONSE TO REVIEWERS – Nature Communications

Klumpp-Thomas et al. 2020

Reviewer #1 (Viral immunity, antibody response.) (Remarks to the Author):

Standardization of enzyme-linked immunosorbent assays for serosurveys of the SARS-CoV-2 pandemic using clinical and at-home blood sampling

The authors have designed and described in detail a novel ELISA protocol for detecting SARS-COV-2 in this paper. The authors have developed a pipeline for producing and validating several SARS-COV-2 antigens from different sources for their ELISA protocol, which will be useful for any labs repeating their ELISA or producing their own ELISA. The authors demonstrated that their ELISA has a high specificity and high sensitivity when tested against their archived negative samples and convalescent blood samples from previously SARS-COV-2 PCR+ individuals. Additionally, the authors showed that their ELISA still works reasonably well for identifying SARS-COV-2 seropositive individuals with samples collected from a hard-hit community. To elucidate if seasonal coronavirus infections could affect their ELISA results, the authors tested OC43, HKU1, SARS-COV-1, and MERS spike proteins and confirmed there was low reactivity in the ELISA. Finally, the authors did a simulation analysis to illustrate how sample size and the various specificities of their ELISA could affect the confidence interval of seroprevalence study.

This manuscript provides a useful protein production and a simple ELISA protocol that can be performed manually or with the assistance of automation. An easy to conduct ELISA assay with high specificity and high sensitivity is useful, as early SARS-COV-2 serology studies have failed to have both specificity and sensitivity. Unfortunately, this ELISA protocol and paper is not per-se novel. ELISA assays are widely available now that have both characteristics- dampening enthusiasm. Moreover, there are a few issues which should be addressed. My critiques are listed below.

Thank you for your feedback and constructive evaluation.

We are aware that other manuscripts on assay platforms have been published since our submission, but we feel that this submission still adds important information to the literature. In particular since this protocol is now adopted and implemented in multiple serosurvey programs, including transNIH longitudinal studies, it is important for the research community to understand the platform and the thorough evaluation that was implemented. Although our research may not have been first to be published, we emphasize that the difference between this protocol and others is the care that was taken to be sure the serosurvey protocol was highly reproducible and to present the results to the scientific community in a comprehensive manuscript.

Comments:

1) The supplementary figures are out of order. Supplementary figure 1 is cited first in the results section, followed by Supplementary Figure 6. Please reorder the supplementary figures so they follow the text.

Thank you, we have re-ordered these.

2) Was specificity and sensitivity calculated from micro sampler elute for Supplementary Figure 2? The data correlates but a few samples deviate in IgM. As this ELISA protocol may be used for seroprevalence studies, which could receive a wide variety of samples; a thorough characterization of a variety of sample inputs would be useful.

We agree that this is a very important point. Sensitivity and specificity are based off of archival (pre-2019) serum samples as micro sampler eluate was collected after the commencement of the pandemic. As we adapted the protocol to minimize in-person contact and allow for use during the rapidly developing

pandemic, we utilized dried blood samples that are broadly accepted as a method of antibody assessment [see references added below]. We examined the correlation of 68 microsample eluate samples to serum samples from the same donor (Supplemental Figure 8).

We have also added two new figures (Figure 5, Supplementary Figure 10) to further examine the similarity of the microsample eluate and serum samples (refer to response to reviewer comment 5 for description of added information, equations of regression lines: Spike IgM $Y = 1.014 * X + 0.04017$, RBD IgM $Y = 0.9949 * X + 0.01501$, Spike IgG $Y = 0.9662 * X + 0.01068$, RBD IgG $Y = 0.9802 * X + 0.02104$). Additionally, to address this directly, we have added the following sentence to the text,

"Though we have displayed data on both serum and dried blood microsamplers (Neoteryx), any adaptation of this protocol further should evaluate each sample source to ensure proper measurements of antibodies in blood or other body fluids."

References:

Bloch, Evan M., et al. "Babesia microti and malaria infection in Africa: a pilot serosurvey in Kilosa District, Tanzania." *The American journal of tropical medicine and hygiene* 99.1 (2018): 51-56.

Pass, Kenneth A., et al. "Comparison of newborn screening records and birth certificates to estimate bias in newborn HIV serosurveys." *American journal of public health* 81.Suppl (1991): 22-24.

Hardelid, P., et al. "Agreement of rubella IgG antibody measured in serum and dried blood spots using two commercial enzyme-linked immunosorbent assays." *Journal of medical virology* 80.2 (2008): 360-364.

Hayford, Kyla, et al. "Measles and rubella serosurvey identifies rubella immunity gap in young adults of childbearing age in Zambia: The added value of nesting a serological survey within a post-campaign coverage evaluation survey." *Vaccine* 37.17 (2019): 2387-2393.

3) The authors state "Overall no correlative effect ($R^2 = 0.3577, 0.2606, 0.2747, \text{ and } 0.4174$, correlation = 0.5981, 0.5105, 0.5241, 0.6461 respectively) was found between SARS-CoV-2 signal intensity and the of the four other coronaviruses (OC43, HKU1, MERS, SARS1). "This statement needs to be changed. No correlation would be $R^2=0$. There is a correlation between the signal intensity of SARS-COV2 and the other coronaviruses.

We agree with the reviewer and have adjusted the statement to the following:

Overall **minimal correlation** ($R^2 = 0.3577, 0.2606, 0.2747, \text{ and } 0.4174$, correlation = 0.5981, 0.5105, 0.5241, 0.6461 respectively) was found between SARS-CoV-2 signal intensity and the signal intensity of the four other coronaviruses (OC43, HKU1, MERS, SARS1). We have researched this phenomenon in depth and found that while **minimal** linear correlative effect is seen, there is potential reactivity between antibodies against SARS-CoV-2 spike protein and other spike proteins, and individuals that express high levels of these cross-reactive antibodies may be of interest for therapeutics.

As we agree that this is an important topic but further analysis is out of the scope of this specific study, we refer the reviewer to our second manuscript building on this data that is currently in revision at another journal and available via preprint: Hicks et al. 2020 Medrxiv. We have added a citation for this preprint in our revised manuscript.

4) Figure 6 has several issues that should be addressed. The utility of the figure is questionable since the text nor the figure make an easily understandable argument for why this analysis demonstrates the usefulness of their specific ELISA in a seroprevalence study. Are the assumptions the simulation makes reasonable to assume for a large scale serology study conducted with this assay? Are 100, 300, or 1000 samples the normal sample size for their serology study? Is a 99% sensitivity expected during a large

scale serology study when the samples come from multiple sources and from patients with a variety of symptoms and dates from infection/symptom onset? In addition, the labeling and description of the red/green/black lines, red/green/black dots, and proportion-axis label is not explained well. Finally, the third graph in Fig6a and b have the green and black lines closer together than the red. This means the black line (100 samples) and green line (1000 samples) agree well but the red line (300 samples) does not, which seems incorrect.

The reviewer is correct. The top right and bottom right plots in figure 6 were incorrect. The sample size in the simulations was inadvertently set to 1000 rather than 100. This has now been corrected. We also agree that the legend is confusing and thus it has been rewritten. The purpose of this analysis is to demonstrate the variability of the prevalence estimates when different numbers of samples are used to estimate the sensitivity and specificity since these estimates are used in the calculation of the estimate of prevalence. When developing the assay we have control over the sensitivity and specificity since we set the cut points. So it is possible to set the cut points so that specificity is very high (this may come at the cost of a lower sensitivity). We also have control of how many samples can be used to develop the cut points. We are currently collecting data on true positives and true negatives that will be used to set the cut points. This figure helps us understand how many of these samples are needed. The assumptions in the figure are reasonable which is why they were chosen for the simulations.

See below for the adjusted figure and text:

Figure 6: Simulation results showing confidence intervals for serosurvey prevalence calculations.

Each graph displays 95% confidence intervals (CI) for the estimate of prevalence from 1000 replications of each condition (including estimating the sensitivity and specificity for each replicate). The CIs are sorted by the lower bound, with lower bounds less than 0 replaced by zero. For all graphs the true sensitivity is 0.90 and the simulations use 100 samples to estimate the sensitivity. For the graphs in row

(a) the true specificity is 0.99 and in row (b) the true specificity is 1.00. The graphs show the simulations results with estimates of specificity using sample sizes of 100 (black), 300 (red), 1000 (green). Points are plotted black, red, then green, so some of the black and green points may be covered (e.g., the lower bounds for all three colors are all zeros in the bottom left panel). The columns give results for true prevalence values of 0.001, 0.01, or 0.1. The true prevalence for each simulation is shown by a vertical gray line.

5) The authors state “This suggests that not only could detection of antibodies be performed reliably, but quantification of antibodies from these mail-in sampling devices was possible (Supplementary Figure 2).” The authors never demonstrate quantifying any of their results. The authors should demonstrate that their assay has a sufficient dynamic range to quantify antibodies in samples. This would further help the paper as a protocol and reagents for quantifying could be included.

The necessary anti-SARS-CoV-2 recombinant antibodies were not available at the original submission of this manuscript to produce standard curves for absolute quantification of antibodies in blood. Since then, we have acquired recombinant antibodies for IgG and IgM and to demonstrate quantification, we have completed a standard curve with these antibodies spiked into seronegative blood both in serum and dried to microsamplers to (1) develop standard curves for quantification and (2) directly compare the quantification between dried blood microsamplers and serum with the same antibody concentration. The following figure has been added to the manuscript:

Figure 5: Quantification of antibody concentration utilizing 4PL sigmoidal model of recombinant antibody spiked into seronegative blood. (a) Anti-RBD recombinant human antibody (IgG and IgM)

was added to whole blood from two seronegative donors, then absorbed to microsamplers and remaining blood was spun down to isolate serum, and analyzed on full spike ectodomain trimer (spike) or receptor binding domain (RBD) ELISA. (b) Direct comparison of absorbance of range of recombinant antibody concentration in serum (y-axis) versus microsampler (x-axis) blood samples. (c) Sigmoidal four parameter logistic (4PL) curve fitting to recombinant antibody dilution series, 95% confidence intervals shown shaded around fit curve. (d) Quantification of IgG levels in a sample high-incidence population. (e) Upper limit of quantification at 1:400 dilution of serum into ELISA (1:10 dilution of microsampler eluate).

a
$$y = a + \frac{x^m(b - a)}{x^m + c^m}$$

b
$$x = c \left(\frac{a - b}{y - b} - 1 \right)^{\frac{1}{m}}$$

c

	Spike IgG	RBD IgG	Spike IgM	RBD IgM
a: Bottom	0.2265	0.1993	0.1801	0.2139
b: Top	4.025	3.988	3.856	4.006
c: IC50	0.0008215	0.0004061	0.002023	0.001046
m: Hill's Slope	2.001	1.942	1.861	2.238

	Range at 1:400 (serum)	1:10 (microsampler eluate)
Threshold LOD/LOQ	0.3006	0.0656
Upper LOQ	159.68	92.521

Supplementary Figure 4: Sigmoidal four parameter logistic curve fitting for quantification of antibody concentrations. (a) Equation for sigmoidal 4PL where y = absorbance, a = minimum ("bottom"), b = maximum ("top"), c = IC50, m = Hill's slope, and x = antibody concentration. (b) Equation solved for x to use in calculating concentration from absorbance. (c) Variables for Spike and RBD IgG and IgM ELISAs, and quantitative range at 1:400 dilution of serum or 1:10 dilution of microsampler eluate. Threshold LOD/LOQ = limit of detection/limit of quantification calculated as the concentration at the determined threshold value for positivity. Upper LOQ = limit of quantification when the instrument reaches saturation of signal and resulting concentrations are at or above the upper LOQ. Sigmoidal 4PL calculated in GraphPad Prism.

And the following text has been added to the manuscript:

In order to further analyze this population and evaluate the potential for quantification, we developed standard curves of recombinant antibodies spiked into control seronegative blood and applied the resulting standard curve to quantify IgG and IgM concentrations (**Figure 5**). Recombinant anti-RBD antibody was spiked into whole blood from two seronegative donors and either directly loaded onto microsamplers or spun down to isolate the serum from spiked blood (**Figure 5a**). The resulting Spike and RBD ELISAs were in agreement between serum and microsamplers with a 1:1 signal ratio as seen previously, suggesting the microsamplers are a valid method of quantifying antibody in blood (**Figure 5b**). A sigmoidal four parameter logistic curve was fit to the resulting optical density values to yield a standard curve of antibody concentration versus OD (**Figure 5c**). Utilizing the resulting equations, we transformed the absorbance values of our small-scale test sample set to concentrations. The lower limits of detection/quantification for our assays is 0.3006 ug/ml (Spike IgG), 0.0656 ug/ml (RBD IgG), 0.1468 ug/ml (Spike IgM), and 0.1609 ug/ml (RBD IgM). The upper limits of quantification at a dilution of 1:400 serum or 1:10 microsampler eluate into the ELISA were: 159.68 ug/ml (Spike IgG), 92.52 ug/ml (RBD IgG), 574 ug/ml (Spike IgM), 116.28 ug/ml (RBD IgM; **Figure 5d**, **Supplementary Figure 4**). Above these concentrations, the detectors on the plate reader are saturated, and samples would need to be titrated down for an exact concentration measurement.

6) Out of the 68 high incidence community, 62 were found to be IgG positive. Out of the remaining 6, all were IgA positive in Supplementary Figure 11k. Should the authors change their strategy from analyzing

IgM and IgG, to IgG and IgA as this seems to detect more positive samples, or is there a potential loss of specificity with IgA?

The reviewer poses a great question. We approached the study without a pre-formed opinion about what the most appropriate protocol/analysis approach would be. We found IgM and IgG measurements to provide a more specific protocol, which is unsurprising given that IgM and IgG are the predominant immunoglobulin forms in blood. IgA-positive signal alone is not indicative of seropositivity. The IgA assays are established and can be assessed in samples received as part of studies, but do not form part of the protocol. The majority of IgA single positives are low positives, coming just above the threshold value. As such, to avoid the potential loss of specificity as the reviewer mentioned, we chose to focus on IgG and IgM as the most convincing and established seropositivity markers. To this extent we have added the following text to the manuscript:

The absorbance (OD) of IgA single positive samples are very close to the threshold level, and no IgA single positive donors were detected with an OD greater than 1. To ensure strong specificity we utilized IgG and IgM as the major antibody classes in the blood serum and utilized IgA to characterize the immune response.

7) Supplementary Fig 7 needs better labels. Are the dots replicates and the x-axis the well positions, or vice versa. Also, were these archival samples or other samples that were tested? Was intermediate precision, day-to-day variability, operator-to-operator variability, laboratory-to-laboratory variability, assessed in addition to repeatability?

Thank you, we have modified this figure. The x-axis represents samples, the dots represent wells. The legend has been updated to reflect that these were archival samples.

Supplementary Figure 3: Technical repeatability from well-to-well and plate-to-plate in IgG and IgM ELISA using a semi-automated setup, analyzing archival negative and convalescent positive control samples.

Parameters such as laboratory-to-laboratory variation were not possible on the timeline described, nor are these published alongside other serology manuscripts and thus it is outside of the scope of the

current manuscript, as there are no current assay methodology papers that provide in-depth operator-to-operator or laboratory-to-laboratory variation studies. We will be publishing all control data alongside our national serosurvey to display operator-to-operator and long-term stability of the assay. Several conclusions made in this manuscript have been repeated by other laboratories, but they are not yet ready to publish their studies.

8) The authors state, "The adaptation of this protocol to analyze both IgM and IgA allows characterization of the stage of infection and increases the sensitivity to identify relatively early-stage infections that have yet to mount a strong IgG response". While it is true that IgM and IgA can be used to characterize the stage of infection in other infections, the authors have provided no data that their assay is sensitive enough to detect an early stage of infection using IgM or IgA before an IgG response has been mounted. This statement should be removed.

We have removed this statement.

9) The authors state "Based on the data presented here, both the manual and semi-automated methods using our protocol provide a sensitivity and specificity of 100% when used both for IgG and IgM." Their data with their archived samples and PCR+ controls have a sensitivity and specificity of 100%, but their hard-hit community data does not have a sensitivity of 100% using IgG and IgM. This statement should be modified to "based on our archived negatives and convalescent blood samples".

The "hard hit community" were not all SARS-CoV-2 positive donors, and thus we do not expect for all to be positive. Only 22 of the 68 were PCR+ the others were known exposure with some symptoms associated with COVID19, but no known diagnosis. It should be re-iterated that the status of the PCR test result was not known for each individual sample. We cannot assume that all 68 were infected with SARS-CoV-2 and thus cannot make any conclusions on sensitivity/specificity utilizing this data set. The "hard hit community" was a test in a high-incidence area with unknown SARS-CoV-2 infection status to mimic our serosurvey. To ensure this is communicated properly we have added the following sentence to our methods section:

This population was a test cohort for our assay, and it was unknown if each individual donor was seropositive or previously infected by SARS-CoV-2 based on a PCR-based detection strategy.

10) A large scale seroprevalence study will encounter many confounding variables such as: severity of infection, duration since infection, duration since symptom onset, age, ethnicity, gender, sample input, sample collection method, time since collection, concurrent non-COVID infections, preexisting conditions, individuals with autoimmune disorders, individuals on immunosuppressants, and more. The authors have not addressed how any of these confounding variables could affect their sensitivity or specificity. This means their specificity and sensitivity could be drastically altered when analyzing large scale sample sets, which could dramatically affect their final results.

As with any clinical study, we envisage that our serosurvey will encounter some of these effects. We look forward to analyzing and sharing the results of our serosurvey, where the participant information described above as confounding variables is included (age, sex, race, ethnicity, presence of symptoms associated with COVID19 and when, travel, occupation, education level, rural/urban, housing type, vaccination history, medical conditions, etc.).

From a sample stability point of view, blood collection and sample handling in our NIH Clinical Center is very standardized. Remote donors are instructed to ship within 24 hours of collection and the samples are shipped overnight back to the NIH. The home sample kits do present interesting challenges, and while not the subject of this manuscript we have simulated a range of scenarios including sampling then leaving on a shelf for longer period to time, high heat (in case of potential shipping delays), cold, physically impacting the device, etc., and not found Ig levels to be impacted. The manufacturer (Neoteryx) has also undertaken similar studies, and other dried blood devices (dried blood spots) have been used regularly in the past for seroprevalence studies, a handful of references are listed below and have been added to the manuscript:

Bloch, Evan M., et al. "Babesia microti and malaria infection in Africa: a pilot serosurvey in Kilosa District, Tanzania." *The American journal of tropical medicine and hygiene* 99.1 (2018): 51-56.

Pass, Kenneth A., et al. "Comparison of newborn screening records and birth certificates to estimate bias in newborn HIV serosurveys." *American journal of public health* 81.Suppl (1991): 22-24.

Hardelid, P., et al. "Agreement of rubella IgG antibody measured in serum and dried blood spots using two commercial enzyme-linked immunosorbent assays." *Journal of medical virology* 80.2 (2008): 360-364.

Hayford, Kyla, et al. "Measles and rubella serosurvey identifies rubella immunity gap in young adults of childbearing age in Zambia: The added value of nesting a serological survey within a post-campaign coverage evaluation survey." *Vaccine* 37.17 (2019): 2387-2393.

Reviewer #2 (Antibody response, Ag mimicry, vaccine.) (Remarks to the Author):

The manuscript by Klumpp-Thomas et al. entitled "Standardization of ..." describes a protocol an ELISA-based to determine thresholds of antibodies that defines seropositivity of serum samples of SARS-CoV-2 patients. The approach described summaries that being conducted by the national serosurvey (sponsored by the NIH) with the intent to standard large cohort studies.

Positive aspects of the manuscript

- 1.) A validated approach to determine immunity and correlates of protection towards and against the virus.
- 2.) Multiple sample types and approaches tested.
- 3.) Good statistical analyses of the results
- 4.) Results support the conclusion drawn.
- 5.) Results observed is similar either by manual or semi-automated procedures.

The manuscript certainly describes the ongoing approach taken in the national Immunosurvey cohort study and well describes the protocol taken. However, the expectation is that this protocol be adopted by multiple labs. Yet the antigens used to develop the protocol may not be readily available and still requires some skill to generate the two antigens that are important to determine the sensitivity described in the paper. Little discussion is directed toward this expectation. Over all the manuscript is well written and provides the technical ground work required for validated analyses of antibodies to the virus spike protein.

We thank the reviewer for their constructive feedback. The detailed protocol for antigen production in this manuscript and our associated manuscript (Esposito et al *Protein Production and Purification* 2020) allows researchers to very accurately reproduce the antigen production to high quality using commercially available reagents. We have produced more than a dozen individual productions of both spike proteins with high levels of reproducibility in both yield and quality (batch-to-batch characterizations are available in the above referenced publication which references this article's preprint). The following citation has been updated in the revised manuscript:

Esposito, D. et al. Optimizing high-yield production of SARS-CoV-2 soluble spike trimers for serology assays. *Protein Expression and Purification*, 105686 (2020).

We are happy to provide guidance to any laboratory that wishes to repeat our protocols and will respond (and have responded) enthusiastically to any individual that reaches out if they have questions.

REVIEWERS' COMMENTS

Reviewer #1 (Remarks to the Author):

all comments have been addressed to the best of the authors ability.